# EXPRESSIVE LOSSES FOR VERIFIED ROBUSTNESS VIA CONVEX COMBINATIONS

**Alessandro De Palma**[1][a]**, Rudy Bunel**[2]**, Krishnamurthy (Dj) Dvijotham**[2]**,**
**M. Pawan Kumar**[2]**, Robert Stanforth**[2]**, Alessio Lomuscio**[3]
[1]Inria, École Normale Supérieure, PSL University, CNRS
[2]Google DeepMind     [3]Imperial College London
alessandro.de-palma@inria.fr

## ABSTRACT

In order to train networks for verified adversarial robustness, it is common to over-approximate the worst-case loss over perturbation regions, resulting in networks that attain verifiability at the expense of standard performance. As shown in recent work, better trade-offs between accuracy and robustness can be obtained by carefully coupling adversarial training with over-approximations. We hypothesize that the expressivity of a loss function, which we formalize as the ability to span a range of trade-offs between lower and upper bounds to the worst-case loss through a single parameter (the over-approximation coefficient), is key to attaining state-of-the-art performance. To support our hypothesis, we show that trivial expressive losses, obtained via convex combinations between adversarial attacks and IBP bounds, yield state-of-the-art results across a variety of settings in spite of their conceptual simplicity. We provide a detailed analysis of the relationship between the over-approximation coefficient and performance profiles across different expressive losses, showing that, while expressivity is essential, better approximations of the worst-case loss are not necessarily linked to superior robustness-accuracy trade-offs.

## 1 INTRODUCTION

In spite of recent and highly-publicized successes (Jumper et al., 2021; Fawzi et al., 2022), serious concerns over the trustworthiness of deep learning systems remain. In particular, the vulnerability of neural networks to adversarial examples (Szegedy et al., 2014; Goodfellow et al., 2015) questions their applicability to safety-critical domains. As a result, many authors devised techniques to formally prove the adversarial robustness of trained neural networks (Lomuscio & Maganti, 2017; Ehlers, 2017; Katz et al., 2017). These algorithms solve a non-convex optimization problem by coupling network over-approximations with search techniques (Bunel et al., 2018). While the use of network verifiers was initially limited to models with hundreds of neurons, the scalability of these tools has steadily increased over the recent years (Bunel et al., 2020a; Botoeva et al., 2020; Henriksen & Lomuscio, 2020; Xu et al., 2021; De Palma et al., 2021c; Wang et al., 2021; Henriksen & Lomuscio, 2021; Ferrari et al., 2022), reaching networks of the size of VGG16 (Simonyan & Zisserman, 2015; Müller et al., 2022). Nevertheless, significant gaps between state-of-the-art verified and empirical robustness persist, the latter referring to specific adversarial attacks. For instance, on ImageNet (Deng et al., 2009), Singh et al. (2023) attain an empirical robustness of $57.70\%$ to $\ell_\infty$ perturbations of radius $\epsilon = 4/255$, whereas the state-of-the-art verified robust accuracy (on a downscaled version of the dataset (Chrabaszcz et al., 2017)) is $9.54\%$ against $\epsilon = 1/255$ (Zhang et al., 2022b). In fact, while empirical robustness can be achieved by training the networks against inexpensive attacks (adversarial training) (Madry et al., 2018; Wong et al., 2020), the resulting networks remain hard to verify.

In order to train networks amenable to formal robustness verification (verified training), a number of works (Mirman et al., 2018; Gowal et al., 2018; Wong & Kolter, 2018; Wong et al., 2018; Zhang et al., 2020; Xu et al., 2020; Shi et al., 2021) directly compute the loss over the network over-approximations that are later used for formal verification (verified loss). The employed approximations are typically

---

[a]Work mostly carried out at Imperial College London, and partly at the University of Oxford.

loose, as the difficulty of the optimization problem associated with training increases with approximation tightness, resulting in losses that are hard to train against (Lee et al., 2021; Jovanović et al., 2022). As a result, while the resulting models are easy to verify, these networks display sub-par standard accuracy. Another line of work enhances adversarial training with verifiability-inducing techniques (Xiao et al., 2019; Balunovic & Vechev, 2020; De Palma et al., 2022): by exploiting stronger verifiers, these algorithms yield better results against small perturbations, but under-perform in other settings. More recently, SABR (Müller et al., 2023), combined the advantages of both strategies by computing the loss on over-approximations of small "boxes" around adversarial attacks and tuning the size of these boxes on each setting, attaining state-of-the-art performance. These results were further improved, at the expense of conceptual simplicity and with increased overhead, by carefully coupling IBP or SABR with latent-space adversarial attacks (Mao et al., 2023). While recent work combine attacks with over-approximations in different ways, they typically seek to accurately represent the exact worst-case loss over perturbation regions (Müller et al., 2023; Mao et al., 2023).

Aiming to provide a novel understanding of the recent verified training literature, we define a class of loss functions, which we call expressive, that range from the adversarial to the verified loss when tuning a parameter $\alpha \in [0, 1]$. We hypothesize that this notion of *expressivity is crucial to produce state-of-the-art trade-offs between accuracy and verified robustness*, and provide support as follows:

- We show that SABR is expressive, and illustrate how other expressive losses can be trivially designed via convex combinations, presenting three examples: (i) CC-IBP, based on combinations between adversarial and over-approximated network outputs within the loss, (ii) MTL-IBP, relying on combinations between the adversarial and verified losses, (iii) Exp-IBP, for which the combination is performed between the logarithms of the losses.

- We present a comprehensive experimental evaluation of CC-IBP, MTL-IBP and Exp-IBP on benchmarks from the robust vision literature, demonstrating that the three methods attain state-of-the-art performance in spite of their conceptual simplicity. We show that a careful tuning of any of the considered expressive losses leads to large improvements, in terms of both standard and verified robust accuracies, upon the best previously-reported results on TinyImageNet and downscaled ImageNet. Our findings point to the importance of expressivity, as opposed to the specific form of the expressive loss function, for verified robustness. Code is available at https://github.com/alessandrodepalma/expressive-losses.

- We analyze the effect of the $\alpha$ parameter on robustness-accuracy trade-offs. Differently from common assumptions in the area, for CC-IBP, MTL-IBP and SABR, better approximations of the worst-case loss do not necessarily correspond to performance improvements.

## 2 BACKGROUND

We employ the following notation: uppercase letters for matrices (for example, $A$), boldface letters for vectors (for example, $\mathbf{a}$), $[\![\cdot]\!]$ to denote integer ranges, brackets for intervals and vector indices ($[\mathbf{a}, \mathbf{b}]$ and $\mathbf{a}[i]$, respectively). Let $f : \mathbb{R}^S \times \mathbb{R}^d \to \mathbb{R}^c$ be a neural network with parameters $\boldsymbol{\theta} \in \mathbb{R}^S$ and accepting $d$-dimensional inputs. Given a dataset $(\mathbf{x}, \mathbf{y}) \sim \mathcal{D}$ with points $\mathbf{x} \in \mathbb{R}^d$ and labels $\mathbf{y} \in \mathbb{R}^l$, the goal of verified training is to obtain a network that satisfies a given property $P : \mathbb{R}^c \times \mathbb{R}^l \to \{0, 1\}$ on $\mathcal{D}$: $(\mathbf{x}, \mathbf{y}) \sim \mathcal{D} \implies P(f(\boldsymbol{\theta}, \mathbf{x}), \mathbf{y})$. We will focus on classification problems, with scalar labels $y \in \mathbb{N}$, and on verifying robustness to adversarial inputs perturbations in $\mathcal{C}(\mathbf{x}, \epsilon) := \{\mathbf{x}' : \|\mathbf{x}' - \mathbf{x}\|_p \leq \epsilon\}$:

$$P(f(\boldsymbol{\theta}, \mathbf{x}), y) = [(\operatorname{argmax}_i f(\boldsymbol{\theta}, \mathbf{x}')[i] = y) \, \forall \, \mathbf{x}' \in \mathcal{C}(\mathbf{x}, \epsilon)] . \tag{1}$$

Property $P(f(\boldsymbol{\theta}, \mathbf{x}), y)$ holds when all the points in a small neighborhood of $\mathbf{x}$ are correctly classified.

### 2.1 ADVERSARIAL TRAINING

Given a loss function $\mathcal{L} : \mathbb{R}^c \times \mathbb{N} \to \mathbb{R}$, a network is typically trained by optimizing $\min_{\boldsymbol{\theta}} \mathbb{E}_{(\mathbf{x}, \mathbf{y}) \in \mathcal{D}} [\mathcal{L}(f(\boldsymbol{\theta}, \mathbf{x}), y)]$ through variants of Stochastic Gradient Descent (SGD). Hence, training for adversarial robustness requires a surrogate loss adequately representing equation (1). Borrowing from the robust optimization literature, Madry et al. (2018) introduce the so-called robust loss:

$$\mathcal{L}^*(f(\boldsymbol{\theta}, \mathbf{x}), y) := \max_{\mathbf{x}' \in \mathcal{C}(\mathbf{x}, \epsilon)} \mathcal{L}(f(\boldsymbol{\theta}, \mathbf{x}'), y) . \tag{2}$$

However, given the non-convexity of the maximization problem, Madry et al. (2018) propose to approximate its computation by a lower bound obtained via an adversarial attack. When $\mathcal{C}(\mathbf{x}, \epsilon)$ is an $\ell_\infty$ ball, a popular attack, PGD, proceeds by taking steps in the direction of the sign of $\nabla_{\mathbf{x}'} \mathcal{L}(f(\boldsymbol{\theta}, \mathbf{x}'), y)$, projecting onto the ball after every iteration. Denoting by $\mathbf{x}_{\text{adv}} \in \mathcal{C}(\mathbf{x}, \epsilon)$ the point returned by the attack, the adversarial loss is defined as follows:

$$\mathcal{L}_{\text{adv}}(f(\boldsymbol{\theta}, \mathbf{x}), y) := \mathcal{L}(f(\boldsymbol{\theta}, \mathbf{x}_{\text{adv}}), y) \leq \mathcal{L}^*(f(\boldsymbol{\theta}, \mathbf{x}), y). \tag{3}$$

Adversarially-trained models typically display strong empirical robustness to adversarial attacks and a reduced natural accuracy compared to standard models. However, as shown in the past (Uesato et al., 2018; Athalye et al., 2018), empirical defenses may break under stronger or different attacks (Croce & Hein, 2020), calling for formal robustness guarantees holding beyond a specific attack scheme. These cannot be obtained in a feasible time on adversarially-trained networks (Xiao et al., 2019). As a result, a number of algorithms have been designed to make networks more amenable to formal verification.

## 2.2 Neural Network Verification

Equation (1) states that a network is provably robust if one can show that all logit differences $\mathbf{z}(\boldsymbol{\theta}, \mathbf{x}, y) := (f(\boldsymbol{\theta}, \mathbf{x})[y] \mathbf{1} - f(\boldsymbol{\theta}, \mathbf{x})) \in \mathbb{R}^c$, except for the ground truth, are positive $\forall \, \mathbf{x}' \in \mathcal{C}(\mathbf{x}, \epsilon)$. Hence, formal robustness verification amounts to computing the sign of the following problem:

$$\min_{\mathbf{x}'} \quad \min_{i \neq y} \mathbf{z}(\boldsymbol{\theta}, \mathbf{x}', y)[i] \qquad \text{s.t.} \qquad \mathbf{x}' \in \mathcal{C}(\mathbf{x}, \epsilon) \tag{4}$$

Problem (4) is known to be NP-hard (Katz et al., 2017): solving it is as hard as exactly computing the robust loss from equation (2). In incomplete verification, the minimization is carried out over a tractable network over-approximation, yielding local lower bounds $\underline{\mathbf{z}}(\boldsymbol{\theta}, \mathbf{x}, y)$ to the logit differences:

$$\underline{\mathbf{z}}(\boldsymbol{\theta}, \mathbf{x}, y)[i] \leq \min_{\mathbf{x}' \in \mathcal{C}(\mathbf{x}, \epsilon)} \mathbf{z}(\boldsymbol{\theta}, \mathbf{x}', y)[i] \quad \forall \, i \in [\![0, c-1]\!].$$

If $\min_{i \neq y} \underline{\mathbf{z}}(\boldsymbol{\theta}, \mathbf{x}, y)[i] > 0$, then the network is proved to be robust. On the other hand, if $\min_{i \neq y} \underline{\mathbf{z}}(\boldsymbol{\theta}, \mathbf{x}, y)[i] < 0$, problem (4) is left undecided: the tightness of the bound is hence crucial to maximize the number of verified properties across a dataset $(\mathbf{x}, \mathbf{y}) \sim \mathcal{D}$. A popular and inexpensive incomplete verifier, Interval Bound Propagation (IBP) (Gowal et al., 2018; Mirman et al., 2018), can be obtained by applying interval arithmetic (Sunaga, 1958; Hickey et al., 2001) to neural networks (see appendix A). If an exact solution to (the sign of) problem (4) is required, incomplete verifiers can be plugged in into a branch-and-bound framework (Bunel et al., 2018; 2020b). These algorithms provide an answer on any given property (complete verification) by recursively splitting the optimization domain (branching), and computing bounds to the solutions of the sub-problems. Lower bounds are obtained via incomplete verifiers, whereas adversarial attacks provide upper bounds. For ReLU activations and high-dimensional inputs, branching is typically performed implicitly by splitting a ReLU into its two linear pieces De Palma et al. (2021c); Ferrari et al. (2022).

## 2.3 Training for Verified Robustness

The verified training literature commonly employs the following assumption (Wong & Kolter, 2018):

**Assumption 2.1.** The loss function $\mathcal{L}$ is translation-invariant: $\mathcal{L}(-\mathbf{z}(\boldsymbol{\theta}, \mathbf{x}, y), y) = \mathcal{L}(f(\boldsymbol{\theta}, \mathbf{x}), y)$. Furthermore, $\mathcal{L}(-\mathbf{z}(\boldsymbol{\theta}, \mathbf{x}, y), y)$ is monotonic increasing with respect to $-\mathbf{z}(\boldsymbol{\theta}, \mathbf{x}, y)[i]$ if $i \neq y$.

Assumption 2.1 holds for popular loss functions such as cross-entropy. In this case, one can provide an upper bound to the robust loss $\mathcal{L}^*$ via the so-called verified loss $\mathcal{L}_{\text{ver}}(f(\boldsymbol{\theta}, \mathbf{x}), y)$, computed on a lower bound to the logit differences $\underline{\mathbf{z}}(\boldsymbol{\theta}, \mathbf{x}, y)$ obtained from an incomplete verifier (Wong & Kolter, 2018):

$$\mathcal{L}^*(f(\boldsymbol{\theta}, \mathbf{x}), y) \leq \mathcal{L}_{\text{ver}}(f(\boldsymbol{\theta}, \mathbf{x}), y) := \mathcal{L}(-\underline{\mathbf{z}}(\boldsymbol{\theta}, \mathbf{x}, y), y). \tag{5}$$

A number of verified training algorithms rely on a loss of the form $\mathcal{L}_{\text{ver}}(f(\boldsymbol{\theta}, \mathbf{x}), y)$ above, where bounds $-\underline{\mathbf{z}}(\boldsymbol{\theta}, \mathbf{x}, y)$ are obtained via IBP (Gowal et al., 2018; Shi et al., 2021), inexpensive linear relaxations (Wong & Kolter, 2018), or a mixture of the two (Zhang et al., 2020). IBP training has been extensively studied (Wang et al., 2022; Mao et al., 2024). In order to stabilize training, the radius of the perturbation over which the bounds are computed is gradually increased from 0 to the target $\epsilon$ value (ramp-up). Earlier approaches (Gowal et al., 2018; Zhang et al., 2020) also transition from the natural to the robust loss during ramp-up: $(1 - \kappa) \mathcal{L}(f(\boldsymbol{\theta}, \mathbf{x}), y) + \kappa \mathcal{L}_{\text{ver}}(f(\boldsymbol{\theta}, \mathbf{x}), y)$, with $\kappa$ linearly increasing from 0 to 0.5 or 1. Zhang et al. (2020) may further transition from lower bounds $\underline{\mathbf{z}}(\boldsymbol{\theta}, \mathbf{x}, y)$

partly obtained via linear relaxations (CROWN-IBP) to IBP bounds within $\mathcal{L}_{\text{ver}}(f(\boldsymbol{\theta}, \mathbf{x}), y)$: $(1 - \kappa) \mathcal{L}(f(\boldsymbol{\theta}, \mathbf{x}), y) + \kappa \mathcal{L}(-((1 - \beta) \mathbf{z}_{\text{IBP}}(\boldsymbol{\theta}, \mathbf{x}, y) + \beta \mathbf{z}_{\text{CROWN-IBP}}(\boldsymbol{\theta}, \mathbf{x}, y)), y)$, with $\beta$ linearly decreased from 1 to 0. Shi et al. (2021) removed both transitions and significantly reduced training times by employing BatchNorm (Ioffe & Szegedy, 2015) along with specialized initialization and regularization. These algorithms attain strong verified robustness, verifiable via inexpensive incomplete verifiers, while paying a relatively large price in standard accuracy. Another line of research seeks to improve standard accuracy by relying on stronger verifiers and coupling adversarial training (§2.1) with verifiability-inducing techniques. Examples include maximizing network local linearity (Xiao et al., 2019), performing the attacks in the latent space over network over-approximations (COLT) (Balunovic & Vechev, 2020), minimizing the area of over-approximations used at verification time while attacking over larger input regions (IBP-R) (De Palma et al., 2022), and optimizing over the sum of the adversarial and IBP losses through a specialized optimizer (Fan & Li, 2021). In most cases $\ell_1$ regularization was found to be beneficial. A more recent work, named SABR (Müller et al., 2023), proposes to compute bounds via IBP over a parametrized subset of the input domain that includes an adversarial attack. Let us denote by $\text{Proj}(\mathbf{a}, \mathcal{A})$ the Euclidean projection of $\mathbf{a}$ on set $\mathcal{A}$. Given $\mathbf{x}_\lambda := \text{Proj}(\mathbf{x}_{\text{adv}}, \mathcal{C}(\mathbf{x}, \epsilon - \lambda\epsilon))$, SABR relies on the following loss:

$$\mathcal{L}(-\underline{\mathbf{z}}_\lambda(\boldsymbol{\theta}, \mathbf{x}, y), y), \text{ where } \underline{\mathbf{z}}_\lambda(\boldsymbol{\theta}, \mathbf{x}, y) \leq \mathbf{z}(\boldsymbol{\theta}, \mathbf{x}', y) \; \forall \mathbf{x}' \in \mathcal{C}(\mathbf{x}_\lambda, \lambda\epsilon) \subseteq \mathcal{C}(\mathbf{x}, \epsilon). \quad (6)$$

The SABR loss is not necessarily an upper bound for $\mathcal{L}^*(f(\boldsymbol{\theta}, \mathbf{x}), y)$. However, it continuously interpolates between $\mathcal{L}(f(\boldsymbol{\theta}, \mathbf{x}_{\text{adv}}), y)$, which it matches when $\lambda = 0$, and $\mathcal{L}_{\text{ver}}(f(\boldsymbol{\theta}, \mathbf{x}), y)$, matched when $\lambda = 1$. By tuning $\lambda \in [0, 1]$, SABR attains state-of-the-art performance under complete verifiers (Müller et al., 2023). The recent STAPS (Mao et al., 2023), combines SABR with improvements on COLT, further improving its performance on some benchmarks at the cost of added complexity.

## 3 LOSS EXPRESSIVITY FOR VERIFIED TRAINING

As seen in §2.3, state-of-the-art verified training algorithms rely on coupling adversarial attacks with over-approximations. Despite this commonality, there is considerable variety in terms of how these are combined, leading to a proliferation of heuristics. We now outline a simple property, which we call *expressivity*, that we will show to be key to effective verified training schemes:

**Definition 3.1.** A parametrized family of losses $\mathcal{L}_\alpha(\boldsymbol{\theta}, \mathbf{x}, y)$ is *expressive* if:

- $\mathcal{L}(f(\boldsymbol{\theta}, \mathbf{x}_{\text{adv}}), y) \leq \mathcal{L}_\alpha(\boldsymbol{\theta}, \mathbf{x}, y) \leq \mathcal{L}_{\text{ver}}(f(\boldsymbol{\theta}, \mathbf{x}), y) \; \forall \alpha \in [0, 1]$;

- $\mathcal{L}_\alpha(\boldsymbol{\theta}, \mathbf{x}, y)$ is continuous and monotonically increasing with respect to $\alpha \in [0, 1]$;

- $\mathcal{L}_0(\boldsymbol{\theta}, \mathbf{x}, y) = \mathcal{L}(f(\boldsymbol{\theta}, \mathbf{x}_{\text{adv}}), y)$;      • $\mathcal{L}_1(\boldsymbol{\theta}, \mathbf{x}, y) = \mathcal{L}_{\text{ver}}(f(\boldsymbol{\theta}, \mathbf{x}), y)$.

Expressive losses can range between the adversarial loss in equation (3) to the verified loss in equation (5) through the value of a single parameter $\alpha \in [0, 1]$, the over-approximation coefficient. As a result, they will be able to span a wide range of trade-offs between verified robustness and standard accuracy. Intuitively, if $\alpha \approx 0$, then $\mathcal{L}_\alpha(\boldsymbol{\theta}, \mathbf{x}, y) << \mathcal{L}^*(f(\boldsymbol{\theta}, \mathbf{x}), y)$, resulting in networks hard to verify even with complete verifiers. If $\alpha \approx 1$, then $\mathcal{L}_\alpha(\boldsymbol{\theta}, \mathbf{x}, y) \approx \mathcal{L}_{\text{ver}}(f(\boldsymbol{\theta}, \mathbf{x}), y)$, producing networks with larger verified robustness when using the incomplete verifier employed to compute $\underline{\mathbf{z}}(\boldsymbol{\theta}, \mathbf{x}, y)$ but with lower standard accuracy. In general, the most effective $\alpha$ will maximize verifiability via the verifier to be employed post-training while preserving as much standard accuracy as possible. The exact value will depend on the efficacy of the verifier and on the difficulty of the verification problem at hand, with larger $\alpha$ values corresponding to less precise verifiers. The loss of SABR from equation (6) satisfies definition 3.1 when setting $\lambda = \alpha$ (see appendix B.4). In order to demonstrate the centrality of expressivity, we now present three minimalistic instantiations of the definition and later show that they yield state-of-the-art results on a variety of benchmarks (§6.1).

## 4 EXPRESSIVITY THROUGH CONVEX COMBINATIONS

We outline three simple ways to obtain expressive losses through convex combinations between adversarial attacks and bounds from incomplete verifiers. Proofs and pseudo-code are provided in appendices B and D, respectively.

### 4.1 CC-IBP

For continuous loss functions like cross-entropy, a family of expressive losses can be easily obtained by taking Convex Combinations (CC) of adversarial and lower bounds to logit differences within $\mathcal{L}$:

$$\mathcal{L}_{\alpha,\text{CC}}(\boldsymbol{\theta}, \mathbf{x}, y) := \mathcal{L}(- \left[ (1 - \alpha) \, \mathbf{z}(\boldsymbol{\theta}, \mathbf{x}_{\text{adv}}, y) + \alpha \, \underline{\mathbf{z}}(\boldsymbol{\theta}, \mathbf{x}, y) \right], y). \tag{7}$$

**Proposition 4.1.** *If $\mathcal{L}(\cdot, y)$ is continuous with respect to its first argument, the parametrized loss $\mathcal{L}_{\alpha,CC}(\boldsymbol{\theta}, \mathbf{x}, y)$ is expressive according to definition 3.1.*

The analytical form of equation (7) when $\mathcal{L}$ is the cross-entropy loss, including a comparison to the loss from SABR, is presented in appendix C. In order to minimize computational costs and to yield favorable optimization problems (Jovanović et al., 2022), we employ IBP to compute $\underline{\mathbf{z}}(\boldsymbol{\theta}, \mathbf{x}, y)$, and refer to the resulting algorithm as CC-IBP. We remark that a similar convex combination was employed in the CROWN-IBP (Zhang et al., 2020) loss (see §2.3). However, while CROWN-IBP takes a combination between IBP and CROWN-IBP lower bounds to $\mathbf{z}(\boldsymbol{\theta}, \mathbf{x}, y)$, we replace the latter by the logit differences associated to an adversarial input $\mathbf{x}_{\text{adv}}$, which is computed via randomly-initialized FGSM (Wong et al., 2020) in most of our experiments (see §6) . Furthermore, rather than increasing the combination coefficient from 0 to 1 during ramp-up, we propose to employ a constant and tunable $\alpha$ throughout training.

### 4.2 MTL-IBP

Convex combinations can be also performed between loss functions $\mathcal{L}$, resulting in:

$$\mathcal{L}_{\alpha,\text{MTL}}(\boldsymbol{\theta}, \mathbf{x}, y) := (1 - \alpha)\mathcal{L}(f(\boldsymbol{\theta}, \mathbf{x}_{\text{adv}}), y) + \alpha \, \mathcal{L}_{\text{ver}}(f(\boldsymbol{\theta}, \mathbf{x}), y). \tag{8}$$

The above loss lends itself to a Multi-Task Learning (MTL) interpretation (Caruana, 1997), with empirical and verified adversarial robustness as the tasks. Under this view, borrowing terminology from multi-objective optimization, $\mathcal{L}_{\alpha,\text{MTL}}(\boldsymbol{\theta}, \mathbf{x}, y)$ amounts to a scalarization of the multi-task problem.

**Proposition 4.2.** *The parametrized loss $\mathcal{L}_{\alpha,MTL}(\boldsymbol{\theta}, \mathbf{x}, y)$ is expressive. Furthermore, if $\mathcal{L}(\cdot, y)$ is convex with respect to its first argument, $\mathcal{L}_{\alpha,CC}(\boldsymbol{\theta}, \mathbf{x}, y) \leq \mathcal{L}_{\alpha,MTL}(\boldsymbol{\theta}, \mathbf{x}, y) \, \forall \, \alpha \in [0, 1]$.*

The second part of proposition 4.2, which applies to cross-entropy, shows that $\mathcal{L}_{\alpha,\text{MTL}}(\boldsymbol{\theta}, \mathbf{x}, y)$ can be seen as an upper bound to $\mathcal{L}_{\alpha,\text{CC}}(\boldsymbol{\theta}, \mathbf{x}, y)$ for $\alpha \in [0, 1]$. Owing to definition 3.1, the bound will be tight when $\alpha \in \{0, 1\}$. Appendix C shows an analytical form for equation (8) when $\mathcal{L}$ is the cross-entropy loss. As in section 4.1, we use IBP to compute $\mathcal{L}_{\text{ver}}(f(\boldsymbol{\theta}, \mathbf{x}), y)$; we refer to the resulting algorithm as MTL-IBP. Given its simplicity, ideas related to equation (8) have previously been adopted in the literature. However, they have never been explored in the context of expressivity and the role of $\alpha$ was never explicitly investigated. Fan & Li (2021) present a loss equivalent to equation (8), treated as a baseline for the proposed algorithm, but only consider $\alpha = 0.5$ and use it within a costly layer-wise optimization scheme similar to COLT (Balunovic & Vechev, 2020). While Fan & Li (2021) propose a specialized multi-task optimizer (AdvIBP), we instead opt for scalarizations, which have recently been shown to yield state-of-the-art results on multi-task benchmarks when appropriately tuned and regularized (Kurin et al., 2022; Xin et al., 2022). Gowal et al. (2018) take a convex combination between the natural (rather than adversarial) loss and the IBP loss; however, instead of tuning a constant $\alpha$, they linearly transition it from 0 to 0.5 during ramp-up (see §2.3). Mirman et al. (2018) employ $\mathcal{L}(f(\boldsymbol{\theta}, \mathbf{x}), y) + 0.1 \, (-\text{softplus}(\min_{i \neq y} \underline{\mathbf{z}}(\boldsymbol{\theta}, \mathbf{x}, y)[i]))$.

### 4.3 EXP-IBP

For strictly positive loss functions $\mathcal{L}$ such as cross-entropy, we can also perform convex combinations between the logarithms of the losses: $\log \mathcal{L}_{\alpha,\text{Exp}}(\boldsymbol{\theta}, \mathbf{x}, y) = (1 - \alpha) \log \mathcal{L}(f(\boldsymbol{\theta}, \mathbf{x}_{\text{adv}}), y) + \alpha \log \mathcal{L}_{\text{ver}}(f(\boldsymbol{\theta}, \mathbf{x}), y)$. By exponentiating both sides, we obtain:

$$\mathcal{L}_{\alpha,\text{Exp}}(\boldsymbol{\theta}, \mathbf{x}, y) := \mathcal{L}(f(\boldsymbol{\theta}, \mathbf{x}_{\text{adv}}), y)^{(1-\alpha)} \, \mathcal{L}_{\text{ver}}(f(\boldsymbol{\theta}, \mathbf{x}), y)^{\alpha}, \tag{9}$$

which is an expressive loss displaying the over-approximation coefficient in the exponents. Analogously to §4.1 and §4.2, we compute $\mathcal{L}_{\text{ver}}(f(\boldsymbol{\theta}, \mathbf{x}), y)$ via IBP, calling the resulting algorithm Exp-IBP.

**Proposition 4.3.** *If $\mathcal{L} : \mathbb{R}^c \times \mathbb{N} \rightarrow \mathbb{R}_{>0}$, the parametrized loss $\mathcal{L}_{\alpha,Exp}(\boldsymbol{\theta}, \mathbf{x}, y)$ is expressive. In addition, $\mathcal{L}_{\alpha,Exp}(\boldsymbol{\theta}, \mathbf{x}, y) \leq \mathcal{L}_{\alpha,MTL}(\boldsymbol{\theta}, \mathbf{x}, y) \, \forall \, \alpha \in [0, 1]$.*

## 5 RELATED WORK

As outlined in §2.3, many verified training algorithms work by either directly upper bounding the robust loss (Gowal et al., 2018; Mirman et al., 2018; Wong & Kolter, 2018; Zhang et al., 2020; Xu et al., 2020; Shi et al., 2021), or by inducing verifiability on top of adversarial training (Xiao et al., 2019; Balunovic & Vechev, 2020; De Palma et al., 2022; Müller et al., 2023; Mao et al., 2023): this work falls in the latter category. Algorithms belonging to these classes are typically employed against $\ell_\infty$ perturbations. Other families of methods include Lipschitz-based regularization for $\ell_2$ perturbations (Leino et al., 2021; Huang et al., 2021), and architectures that are designed to be inherently robust for either $\ell_2$ (Trockman & Kolter, 2021; Singla & Feizi, 2021; Singla et al., 2022; Singla & Feizi, 2022; Xu et al., 2022) or $\ell_\infty$ (Zhang et al., 2021; 2022a) perturbations, for which SortNet (Zhang et al., 2022b) is the most recent and effective algorithm. A prominent line of work has focused on achieving robustness with high probability through randomization (Cohen et al., 2019; Salman et al., 2019): we here focus on deterministic methods.

IBP, presented in §2.2, is arguably the simplest incomplete verifier. A popular class of algorithms replaces the activation by over-approximations corresponding to pairs of linear bound, each respectively providing a lower or upper bound to the activation function (Wong & Kolter, 2018; Singh et al., 2018; Zhang et al., 2018; Singh et al., 2019b). Tighter over-approximations can be obtained by representing the convex hull of the activation function (Ehlers, 2017; Bunel et al., 2020a; Xu et al., 2021), the convex hull of the composition of the activation with the preceding linear layer (Anderson et al., 2020; Tjandraatmadja et al., 2020; De Palma et al., 2021a;b), interactions within the same layer (Singh et al., 2019a; Müller et al., 2022), or by relying on semi-definite relaxations (Raghunathan et al., 2018; Dathathri et al., 2020; Lan et al., 2022). In general, the tighter the bounds, the more expensive their computation. As seen in the recent neural network verification competitions (Bak et al., 2021; Müller et al., 2022), state-of-the-art complete verifiers based on branch-and-bound typically rely on specialized and massively-parallel incomplete verifiers (Wang et al., 2021; Ferrari et al., 2022; Zhang et al., 2022c), paired with custom branching heuristics (Bunel et al., 2020b; De Palma et al., 2021c; Ferrari et al., 2022).

## 6 EXPERIMENTAL EVALUATION

We now present experimental results that support the centrality of expressivity, as defined in §3, to verified training. In §6.1 we compare CC-IBP, MTL-IBP and Exp-IBP with results from the literature, showing that even minimalistic instances of the definition attain state-of-the-art performance on a variety of benchmarks. In §6.2 and §6.3 we study the effect of the over-approximation coefficient on the performance profiles of expressive losses, highlighting their sensitivity to its value and showing that better approximations of the branch-and-bound loss do not necessarily results in better performance.

We implemented CC-IBP, MTL-IBP and Exp-IBP in PyTorch (Paszke et al., 2019), starting from the training pipeline by Shi et al. (2021), based on automatic LiRPA (Xu et al., 2020), and exploiting their specialized initialization and regularization techniques (see appendix F.2). All our experiments use the same 7-layer architecture as previous work (Shi et al., 2021; Müller et al., 2023), which employs BatchNorm (Ioffe & Szegedy, 2015) after every layer. Except on CIFAR-10 with $\epsilon = 2/255$, for which an 8-step attack is used as in previous work (Müller et al., 2023), we employ randomly-initialized single-step attacks (Wong et al., 2020) to compute $\mathbf{x}_{adv}$ and force the points to be on the perturbation boundary via a large step size (see appendices G.9 and G.4 for ablations on the step size and employed adversarial attack, respectively). Unless specified otherwise, the reported verified robust accuracies for our experiments are computed using branch-and-bound, with a setup similar to IBP-R (De Palma et al., 2022). Specifically, we use the OVAL framework (Bunel et al., 2020b; De Palma et al., 2021c), with a variant of the UPB branching strategy (De Palma et al., 2022) and $\alpha$-$\beta$-CROWN as the bounding algorithm (Wang et al., 2021) (see appendix E). Details pertaining to the employed datasets, hyper-parameters, network architectures, and computational setup are reported in appendix F.

### 6.1 COMPARISON WITH LITERATURE RESULTS

Table 1 compares the results of CC-IBP, MTL-IBP and Exp-IBP with relevant previous work in terms of standard and verified robust accuracy against $\ell_\infty$ norm perturbations. We benchmark on the CIFAR-10 (Krizhevsky & Hinton, 2009), TinyImageNet (Le & Yang, 2015), and downscaled ImageNet

Table 1: Comparison of the expressive losses from §4 with literature results for $\ell_\infty$ norm perturbations on CIFAR-10, TinyImageNet and downscaled $(64 \times 64)$ ImageNet. The entries corresponding to the best standard or verified robust accuracy for each perturbation radius are highlighted in bold. For the remaining entries, improvements on the literature performance of the best-performing ReLU-based architectures are underlined.

| Dataset | $\epsilon$ | Method | Source | Standard acc. [%] | Verified rob. acc. [%] |
|---|---|---|---|---|---|
| CIFAR-10 | $\frac{2}{255}$ | CC-IBP | this work | 80.09 | **63.78** |
| | | MTL-IBP | this work | 80.11 | 63.24 |
| | | Exp-IBP | this work | **80.61** | 61.65 |
| | | STAPS | Mao et al. (2023) | 79.76 | 62.98 |
| | | SABR[†] | Mao et al. (2024) | 79.89 | 63.28 |
| | | SortNet | Zhang et al. (2022b) | 67.72 | 56.94 |
| | | IBP-R[†] | Mao et al. (2024) | 80.46 | 62.03 |
| | | IBP[†] | Mao et al. (2024) | 68.06 | 56.18 |
| | | AdvIBP | Fan & Li (2021) | 59.39 | 48.34 |
| | | CROWN-IBP | Zhang et al. (2020) | 71.52 | 53.97 |
| | | COLT | Balunovic & Vechev (2020) | 78.4 | 60.5 |
| | $\frac{8}{255}$ | CC-IBP | this work | 53.71 | 35.27 |
| | | MTL-IBP | this work | 53.35 | 35.44 |
| | | Exp-IBP | this work | 53.97 | 35.04 |
| | | STAPS | Mao et al. (2023) | 52.82 | 34.65 |
| | | SABR | Müller et al. (2023) | 52.38 | 35.13 |
| | | SortNet | Zhang et al. (2022b) | **54.84** | **40.39** |
| | | IBP-R | De Palma et al. (2022) | 52.74 | 27.55 |
| | | IBP | Shi et al. (2021) | 48.94 | 34.97 |
| | | AdvIBP | Fan & Li (2021) | 47.14 | 33.43 |
| | | CROWN-IBP | Xu et al. (2020) | 46.29 | 33.38 |
| | | COLT | Balunovic & Vechev (2020) | 51.70 | 27.50 |
| TinyImageNet | $\frac{1}{255}$ | CC-IBP | this work | 38.61 | **26.39** |
| | | MTL-IBP | this work | 37.56 | 26.09 |
| | | Exp-IBP | this work | **38.71** | 26.18 |
| | | STAPS | Mao et al. (2023) | 28.98 | 22.16 |
| | | SABR[†] | Mao et al. (2024) | 28.97 | 21.36 |
| | | SortNet | Zhang et al. (2022b) | 25.69 | 18.18 |
| | | IBP[†] | Mao et al. (2024) | 25.40 | 19.92 |
| | | CROWN-IBP | Shi et al. (2021) | 25.62 | 17.93 |
| ImageNet64 | $\frac{1}{255}$ | CC-IBP | this work | 19.62 | 11.87 |
| | | MTL-IBP | this work | 20.15 | 12.13 |
| | | Exp-IBP | this work | **22.73** | **13.30** |
| | | SortNet | Zhang et al. (2022b) | 14.79 | 9.54 |
| | | CROWN-IBP | Xu et al. (2020) | 16.23 | 8.73 |
| | | IBP | Gowal et al. (2018) | 15.96 | 6.13 |

[†] $2\times$ wider network than the architecture used in our experiments and larger than those from the relevant works: Shi et al. (2021) for IBP, De Palma et al. (2022) for IBP-R, Müller et al. (2023) for SABR.

$(64\times64)$ (Chrabaszcz et al., 2017) datasets. The results from the literature report the best performance attained by each algorithm on any architecture, thus providing a summary of the state-of-the-art. In order to produce a fair comparison with literature results, in this section we comply with the seemingly standard practice in the area (Gowal et al., 2018; Zhang et al., 2020; Shi et al., 2021; Müller et al., 2023) and directly tune on the evaluation sets. Table 1 shows that, in terms of robustness-accuracy trade-offs, CC-IBP, MTL-IBP and Exp-IBP all attain state-of-the-art performance. In particular, no literature results correspond to better trade-offs in any of the considered benchmarks except for CIFAR-10 with $\epsilon = 8/255$, where specialized architectures such as SortNet attain better robustness-accuracy trade-offs. On larger datasets, our Exp-IBP, MTL-IBP and CC-IBP experiments display significant improvements upon the results from the literature. For TinyImageNet, standard and verified robust accuracies are increased by at least $8.58\%$ and $3.93\%$, respectively. On downscaled ImageNet, the improvements correspond to at least $3.39\%$ for standard accuracy and to at least $2.33\%$ for verified robust accuracy. In order to further investigate the role of expressivity in these results, we performed our own tuning of SABR (Müller et al., 2023), which satisfies definition 3.1, on TinyImageNet and ImageNet64 and present the relative results in appendix G.6. Table 9 shows that, when carefully tuned, all the considered expressive losses (CC-IBP, MTL-IBP, Exp-IBP and SABR) attain relatively similar

performance profiles and markedly improve upon previously-reported TinyImageNet and ImageNet64 results. Given the costs associated with ImageNet64, we only attempted branch-and-bound based verification on networks with $\alpha \geq 5 \times 10^{-2}$ for all expressive losses, skewing the performance in favor of Exp-IBP owing to its lower loss values (see proposition 4.3 and table 3). We speculate that better trade-offs in this setup may be obtained when employing lower coefficients. In addition, MTL-IBP performs significantly better than the closely-related AdvIBP (see §4.2) on all benchmarks, confirming the effectiveness of an appropriately tuned scalarization (Xin et al., 2022). Fan & Li (2021) provide results for the use of $\mathcal{L}_{0.5,\text{MTL}}(\boldsymbol{\theta}, \mathbf{x}, y)$ within a COLT-like stage-wise training procedure, treated as a baseline, on CIFAR-10 with $\epsilon = 8/255$, displaying $26.22\%$ IBP verified robust accuracy. These results are markedly worse than those we report for MTL-IBP, which attains $34.61\%$ IBP verified robust accuracy with $\alpha = 0.5$ in table 6 (see appendix F). We believe this discrepancy to be linked to the commonalities with COLT, which under-performs in this setting, and, in line with similar results in the multi-task literature (Kurin et al., 2022), to our use of specialized initialization and regularization from Shi et al. (2021). Additional experimental data is presented in the appendices: appendix G.2 provides an indication of training cost, appendix G.1 presents MNIST results, and appendix G.3 displays verified robust accuracies under less expensive verifiers. Remarkably, on TinyImageNet and downscaled ImageNet, CC-IBP, MTL-IBP and Exp-IBP all attain better verified robust accuracy than literature results even without resorting to complete verifiers.

## 6.2 SENSITIVITY TO OVER-APPROXIMATION COEFFICIENT

Figure 1 reports standard, adversarial, and verified robust accuracies under different verifiers when varying the value of $\alpha$ for CC-IBP, MTL-IBP and Exp-IBP. BaB denotes verified accuracy under the complete verifier from table 1, while CROWN/IBP the one under the best bounds between full backward CROWN and IBP. Adversarial robustness is computed against the attacks performed within the employed BaB framework (see appendix E). As expected, the standard accuracy steadily decreases with increasing $\alpha$ for both algorithms and, generally speaking (except for MTL-IBP and Exp-IBP when $\epsilon = 8/255$), the IBP verified accuracy will increase with $\alpha$. As seen on SABR (Müller et al., 2023), the behavior of the adversarial and verified robust accuracies under tighter verifiers is less straightforward. In fact, depending on the setting and the algorithm, a given accuracy may first increase and then decrease with $\alpha$. This sensitivity highlights the need for careful tuning according to the desired robustness-accuracy trade-off and verification budget. For instance, figure 1 suggests that, for $\epsilon = 8/255$, the best trade-offs between BaB verified accuracy and standard accuracy lie around $\alpha = 0.5$ for both CC-IBP, MTL-IBP and Exp-IBP. When $\epsilon = 2/255$, this will be around $\alpha = 10^{-2}$ for CC-IBP, and when $\alpha \in (10^{-3}, 10^{-2})$ for MTL-IBP, whose natural accuracy decreases more sharply with $\alpha$ (see appendix F). Exp-IBP is extremely sensitive to $\alpha$ on this setup, with the best trade-offs between BaB-verified robustness and standard accuracy lying in a small neighborhood of $\alpha = 10^{-1}$. Note that, as expected from proposition 4.3, Exp-IBP requires larger values of $\alpha$ than MTL-IBP in order attain similar performance profiles. Furthermore, larger $\alpha$ values appear to be beneficial against larger perturbations, where branch-and-bound is less effective. Appendix G.7 carries out a similar analysis on the loss values associated to the same networks.

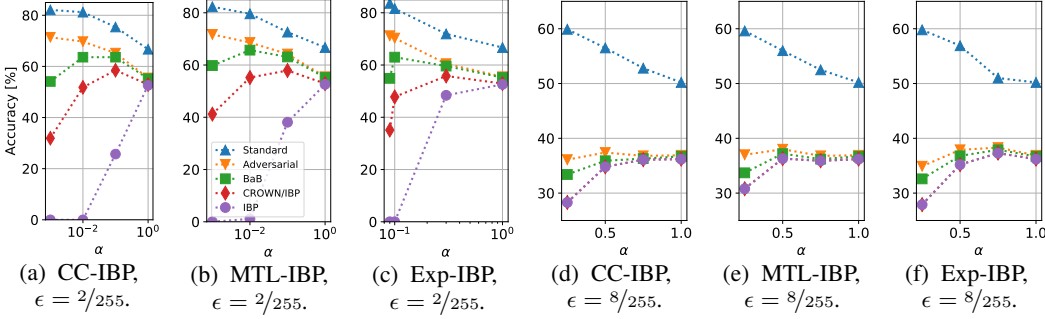

(a) CC-IBP, $\epsilon = 2/255$.    (b) MTL-IBP, $\epsilon = 2/255$.    (c) Exp-IBP, $\epsilon = 2/255$.    (d) CC-IBP, $\epsilon = 8/255$.    (e) MTL-IBP, $\epsilon = 8/255$.    (f) Exp-IBP, $\epsilon = 8/255$.

Figure 1: Sensitivity of CC-IBP, MTL-IBP and Exp-IBP to the convex combination coefficient $\alpha$. We report standard, adversarial and verified robust accuracies (with different verifiers) under $\ell_\infty$ perturbations on the first 1000 images of the CIFAR-10 test set. The legend in plot 1(b) applies to all sub-figures.

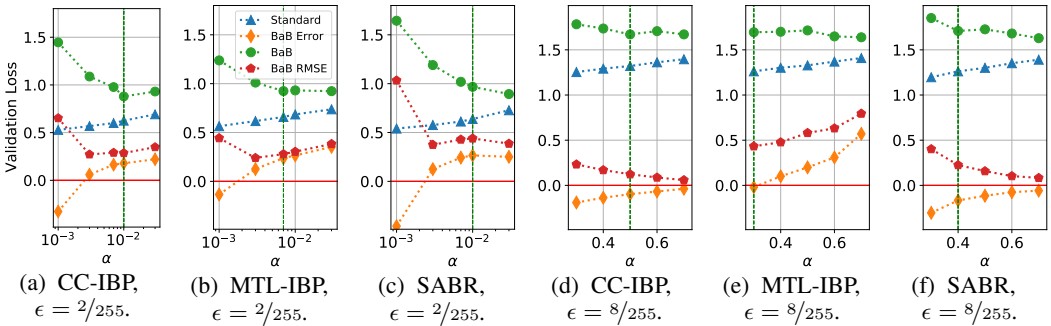

Figure 2: Relationship between the over-approximation coefficient $\alpha$ and the standard loss, the branch-and-bound loss, its approximation error and RMSE (relative to the expressive loss employed during training) on a holdout validation set of CIFAR-10. While the standard loss is computed on the entire validation set, branch-and-bound-related statistics are limited to the first 100 images. The dashed vertical line denotes the $\alpha$ value minimizing the sum of the standard and branch-and-bound losses. The legend in plot 2(b) applies to all sub-figures.

## 6.3 BRANCH-AND-BOUND LOSS AND APPROXIMATIONS

A common assumption in previous work is that better approximations of the worst-case loss from equation (2) will result in better trade-offs between verified robustness and accuracy (Müller et al., 2023; Mao et al., 2023). In order to investigate the link between expressive losses, the worst-case loss, and a principled tuning process, we conduct an in-depth study on a CIFAR-10 validation set for CC-IBP, MTL-IBP and SABR. Specifically, we evaluate on a holdout set taken from the last 20% of the CIFAR-10 training images, and train the same architecture employed in §6.1 on the remaining 80%, tuning $\alpha \in \{1 \times 10^{-3}, 3 \times 10^{-3}, 7 \times 10^{-3}, 1 \times 10^{-2}, 3 \times 10^{-2}\}$ for $\epsilon = 2/255$ and $\alpha \in \{0.3, 0.4, 0.5, 0.6, 0.7\}$ for $\epsilon = 8/255$. Given that computing the worst-case loss on networks of practical interest is intractable, we replace it by $\mathcal{L}_{\text{BaB}}(f(\boldsymbol{\theta}, \mathbf{x}_i), y)$, computed by running branch-and-bound (BaB) for 15 seconds on each logit difference, without returning early for verified properties so as to tighten the bound. Figure 2 reports values for the standard loss, the branch-and-bound loss and two measures for its approximation: the average value of $(\mathcal{L}_\alpha(\boldsymbol{\theta}, \mathbf{x}_i, y) - \mathcal{L}_{\text{BaB}}(f(\boldsymbol{\theta}, \mathbf{x}_i), y))$ (BaB Error) and the square root of the average of its square (BaB RMSE). Observe that the best-performing model, here defined as the one with the smallest sum of the standard and BaB losses, does not necessarily correspond to the points with the best BaB loss approximation. In particular, over-approximations (BaB Error > 0) display better validation trade-offs for $\epsilon = 2/255$, and under-approximations for $\epsilon = 8/255$. Given that $\mathcal{L}_{\text{BaB}}(f(\boldsymbol{\theta}, \mathbf{x}), y) \geq \mathcal{L}^*(f(\boldsymbol{\theta}, \mathbf{x}), y)$, the results on $\epsilon = 2/255$ demonstrate that accurate worst-case loss approximations do not necessarily correspond to better performance: neither in terms of verified robustness alone, nor in terms of trade-offs with accuracy. As shown in appendix G.5, the selected $\alpha$ values result in similar (state-of-the-art for ReLU networks) CIFAR-10 robustness-accuracy trade-offs across the three expressive losses here considered, suggesting the importance of expressivity, rather than of the specific employed heuristic, for verified training.

## 7 CONCLUSIONS

We defined a class of parametrized loss functions, which we call expressive, that range from the adversarial loss to the IBP loss by means of a single parameter $\alpha \in [0, 1]$. We argued that expressivity is key to maximize trade-offs between verified robustness and standard accuracy, and supported this claim by extensive experimental results. In particular, we showed that SABR is expressive, and demonstrated that even three minimalistic instantiations of the definition obtained through convex combinations, CC-IBP, MTL-IBP and Exp-IBP, attain state-of-the-art results on a variety of benchmarks. Furthermore, we provided a detailed analysis of the influence of the over-approximation coefficient $\alpha$ on the properties of networks trained via expressive losses, demonstrating that, differently from what previously conjectured, better approximations of the branch-and-bound loss do not necessarily result in better performance.

## ETHICS STATEMENT

We do not foresee any immediate negative application of networks that are provably robust to adversarial perturbations. Nevertheless, the development of systems that are more robust may hinder any positive use of adversarial examples (Albert et al., 2021), potentially amplifying the harm caused by unethical machine learning systems. On the other hand, the benefits of provable robustness include defending ethical machine learning systems against unethical attackers. Finally, while we demonstrated state-of-the-art performance on a suite of commonly-employed vision datasets, this does not guarantee the effectiveness of our methods beyond the tested benchmarks and, in particular, on different data domains.

## REPRODUCIBILITY STATEMENT

Code to reproduce our experiments and relevant trained models are available at `https://github.com/alessandrodepalma/expressive-losses`. Pseudo-code for both losses can be found in appendix D. Training details, hyper-parameters and computational setups are provided in sections 4, 6 and appendix F. Appendix G.8 provides an indication of experimental variability for MTL-IBP and CC-IBP by repeating the experiments for a single benchmark a further 3 times.

### ACKNOWLEDGMENTS

The authors would like to thank the anonymous reviewers for their constructive feedback, which strengthened the work, and Yuhao Mao, who led us to a correction of the description of our implementation details in appendix F.2. This research was partly funded by UKRI ("SAIS: Secure AI AssistantS", EP/T026731/1) and Innovate UK (EU Underwrite project "EVENFLOW", 10042508). The work was further supported by the "SAIF" project, funded by the "France 2030" government investment plan managed by the French National Research Agency, under the reference ANR-23-PEIA-0006. While at the University of Oxford, ADP was supported by the EPSRC Centre for Doctoral Training in Autonomous Intelligent Machines and Systems, grant EP/L015987/1, and an IBM PhD fellowship. AL is supported by a Royal Academy of Engineering Fellowship in Emerging Technologies.

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

## A  Interval Bound Propagation

As outlined in §2.2, incomplete verification aims to compute lower bounds $\underline{\mathbf{z}}(\boldsymbol{\theta}, \mathbf{x}, y)$ to the logit differences $\mathbf{z}(\boldsymbol{\theta}, \mathbf{x}, y) := (f(\boldsymbol{\theta}, \mathbf{x})[y] \mathbf{1} - f(\boldsymbol{\theta}, \mathbf{x}))$ of network $f(\boldsymbol{\theta}, \mathbf{x})$ for the perturbation domain $\mathcal{C}(\mathbf{x}, \epsilon)$. In the following, we will assume that the last layer of $f(\boldsymbol{\theta}, \mathbf{x})$ and the difference in the definition of $\mathbf{z}(\boldsymbol{\theta}, \mathbf{x}, y)$ have been merged into a single linear layer, representing $\mathbf{z}(\boldsymbol{\theta}, \mathbf{x}, y)$ as an $n$-layer network. We will henceforth use $[\mathbf{a}]_+ := \max(\mathbf{a}, \mathbf{0})$ and $[\mathbf{a}]_- := \min(\mathbf{a}, \mathbf{0})$. Furthermore, let $W_k$ and $\mathbf{b}_k$ be respectively the weight matrix and the bias of the $k$-th layer of $\mathbf{z}(\boldsymbol{\theta}, \mathbf{x}, y)$. In the case of monotonically increasing activation functions $\sigma$, such as ReLU, IBP computes $\underline{\mathbf{z}}(\boldsymbol{\theta}, \mathbf{x}, y)$ as $\hat{\mathbf{l}}_n$ from the following iterative procedure:

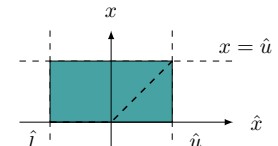

Figure 3: IBP ReLU over-approximation, with $\hat{x} \in [\hat{l}, \hat{u}]$ for the considered input domain.

$$
\begin{aligned}
\hat{\mathbf{l}}_1 &= \min_{\mathbf{x}' \in \mathcal{C}(\mathbf{x}, \epsilon)} W_1 \mathbf{x}' + \mathbf{b}_1 \\
\hat{\mathbf{u}}_1 &= \max_{\mathbf{x}' \in \mathcal{C}(\mathbf{x}, \epsilon)} W_1 \mathbf{x}' + \mathbf{b}_1
\end{aligned}
\quad
\left\{
\begin{aligned}
\hat{\mathbf{u}}_k &= [W_k]_+ \, \sigma(\hat{\mathbf{u}}_{k-1}) + [W_k]_- \sigma(\hat{\mathbf{l}}_{k-1}) + \mathbf{b}_k \\
\hat{\mathbf{l}}_k &= [W_k]_+ \, \sigma(\hat{\mathbf{l}}_{k-1}) + [W_k]_- \sigma(\hat{\mathbf{u}}_{k-1}) + \mathbf{b}_k
\end{aligned}
\right\}
\; k \in [\![2, n]\!].
\tag{10}
$$

For $\ell_\infty$ perturbations of radius $\epsilon$, $\mathcal{C}(\mathbf{x}, \epsilon) = \{\mathbf{x}' : \|\mathbf{x}' - \mathbf{x}\|_\infty \le \epsilon\}$: the linear minimization and maximization oracles evaluate to $\hat{\mathbf{u}}_1 = W_1 \mathbf{x} + \epsilon |W_1| \mathbf{1} + \mathbf{b}_1$ and $\hat{\mathbf{l}}_1 = W_1 \mathbf{x} - \epsilon |W_1| \mathbf{1} + \mathbf{b}_1$, respectively. IBP corresponds to solving an approximation of problem (4) where all the network layers are replaced by their bounding boxes. For instance, given some $\hat{l}$ and $\hat{u}$ from the above procedure, the ReLUs are replaced by the over-approximation depicted in figure 3. Assuming $[W_k]_+$ and $[W_k]_-$ have been pre-computed for $k \in [\![2, n]\!]$, the cost of computing $\underline{\mathbf{z}}(\boldsymbol{\theta}, \mathbf{x}, y)$ via IBP roughly corresponds to four network evaluations. This can be further reduced to roughly two network evaluations (pre-computing $|W_k|$ for $k \in [\![2, n]\!]$) by using $\hat{\mathbf{u}}_k = (\hat{\mathbf{u}}_k + \hat{\mathbf{l}}_k)/2 + (\hat{\mathbf{u}}_k - \hat{\mathbf{l}}_k)/2$ and $\hat{\mathbf{l}}_k = (\hat{\mathbf{u}}_k + \hat{\mathbf{l}}_k)/2 - (\hat{\mathbf{u}}_k - \hat{\mathbf{l}}_k)/2$.

## B  Expressivity Proofs

We here provide the proofs of the expressivity of CC-IBP (appendix B.1), MTL-IBP (appendix B.2), Exp-IBP (appendix B.3), and SABR (appendix B.4).

### B.1  CC-IBP

**Proposition 4.1.** *If $\mathcal{L}(\cdot, y)$ is continuous with respect to its first argument, the parametrized loss $\mathcal{L}_{\alpha, CC}(\boldsymbol{\theta}, \mathbf{x}, y)$ is expressive according to definition 3.1.*

*Proof.* Let us recall that $\mathcal{L}_{\alpha, \mathrm{CC}}(\boldsymbol{\theta}, \mathbf{x}, y) := \mathcal{L}(-[(1 - \alpha)\,\mathbf{z}(\boldsymbol{\theta}, \mathbf{x}_{\mathrm{adv}}, y) + \alpha\,\underline{\mathbf{z}}(\boldsymbol{\theta}, \mathbf{x}, y)], y)$. Trivially, $\mathcal{L}_{0, \mathrm{CC}}(\boldsymbol{\theta}, \mathbf{x}, y) = \mathcal{L}(f(\boldsymbol{\theta}, \mathbf{x}_{\mathrm{adv}}), y)$, and $\mathcal{L}_{1, \mathrm{CC}}(\boldsymbol{\theta}, \mathbf{x}, y) = \mathcal{L}_{\mathrm{ver}}(f(\boldsymbol{\theta}, \mathbf{x}), y)$. By definition, $\underline{\mathbf{z}}(\boldsymbol{\theta}, \mathbf{x}, y) \le \mathbf{z}(\boldsymbol{\theta}, \mathbf{x}_{\mathrm{adv}}, y)$, as $\mathbf{x}_{\mathrm{adv}} \in \mathcal{C}(\mathbf{x}, \epsilon)$. Hence,

$$
\underline{\mathbf{z}}(\boldsymbol{\theta}, \mathbf{x}, y) \le (1 - \alpha)\,\mathbf{z}(\boldsymbol{\theta}, \mathbf{x}_{\mathrm{adv}}, y) + \alpha\,\underline{\mathbf{z}}(\boldsymbol{\theta}, \mathbf{x}, y) \le \mathbf{z}(\boldsymbol{\theta}, \mathbf{x}_{\mathrm{adv}}, y) \text{ for } \alpha \in [0, 1].
$$

Then, owing to assumption 2.1 and to $\mathbf{z}(\boldsymbol{\theta}, \mathbf{x}_{\mathrm{adv}}, y)[y] = \underline{\mathbf{z}}(\boldsymbol{\theta}, \mathbf{x}, y)[y] = 0$, we can apply the loss function $\mathcal{L}(\cdot, y)$ while leaving the inequalities unchanged, yielding:

$$
\mathcal{L}(f(\boldsymbol{\theta}, \mathbf{x}_{\mathrm{adv}}), y) \le \mathcal{L}_\alpha(\boldsymbol{\theta}, \mathbf{x}, y) \le \mathcal{L}_{\mathrm{ver}}(f(\boldsymbol{\theta}, \mathbf{x}), y).
$$

Finally, if $\mathcal{L}(\cdot, y)$ is continuous with respect to its first argument, then $\mathcal{L}(-[(1 - \alpha)\,\mathbf{z}(\boldsymbol{\theta}, \mathbf{x}_{\mathrm{adv}}, y) + \alpha\,\underline{\mathbf{z}}(\boldsymbol{\theta}, \mathbf{x}, y)], y)$ is continuous with respect to $\alpha$ (composition of continuous functions), which concludes the proof. $\qquad\square$

### B.2  MTL-IBP

**Proposition 4.2.** *The parametrized loss $\mathcal{L}_{\alpha, MTL}(\boldsymbol{\theta}, \mathbf{x}, y)$ is expressive. Furthermore, if $\mathcal{L}(\cdot, y)$ is convex with respect to its first argument, $\mathcal{L}_{\alpha, CC}(\boldsymbol{\theta}, \mathbf{x}, y) \le \mathcal{L}_{\alpha, MTL}(\boldsymbol{\theta}, \mathbf{x}, y) \,\forall\, \alpha \in [0, 1]$.*

*Proof.* We start by recalling that $\mathcal{L}_{\alpha,\text{MTL}}(\boldsymbol{\theta}, \mathbf{x}, y) := (1-\alpha)\mathcal{L}(f(\boldsymbol{\theta}, \mathbf{x}_{\text{adv}}), y) + \alpha\,\mathcal{L}_{\text{ver}}(f(\boldsymbol{\theta}, \mathbf{x}), y)$. The proof for the first part of the statement closely follows the proof for proposition 4.1, while not requiring assumption 2.1 nor continuity of the loss function.

The inequality $\mathcal{L}_{\alpha,\text{CC}}(\boldsymbol{\theta}, \mathbf{x}, y) \leq \mathcal{L}_{\alpha,\text{MTL}}(\boldsymbol{\theta}, \mathbf{x}, y)\ \forall\,\alpha \in [0,1]$ trivially follows from the definition of a convex function (Boyd & Vandenberghe, 2004, section 3.1.1) and equation (5). $\qquad\square$

### B.3 EXP-IBP

**Proposition 4.3.** *If $\mathcal{L}: \mathbb{R}^c \times \mathbb{N} \to \mathbb{R}_{>0}$, the parametrized loss $\mathcal{L}_{\alpha,\text{Exp}}(\boldsymbol{\theta}, \mathbf{x}, y)$ is expressive. In addition, $\mathcal{L}_{\alpha,\text{Exp}}(\boldsymbol{\theta}, \mathbf{x}, y) \leq \mathcal{L}_{\alpha,\text{MTL}}(\boldsymbol{\theta}, \mathbf{x}, y)\ \forall\,\alpha \in [0,1]$.*

*Proof.* Recall that $\mathcal{L}_{\alpha,\text{Exp}}(\boldsymbol{\theta}, \mathbf{x}, y) := \mathcal{L}(f(\boldsymbol{\theta}, \mathbf{x}_{\text{adv}}), y)^{(1-\alpha)}\,\mathcal{L}_{\text{ver}}(f(\boldsymbol{\theta}, \mathbf{x}), y)^{\alpha}$. It is easy to see that $\mathcal{L}_{0,\text{Exp}}(\boldsymbol{\theta}, \mathbf{x}, y) = \mathcal{L}(f(\boldsymbol{\theta}, \mathbf{x}_{\text{adv}}), y)$, and $\mathcal{L}_{1,\text{Exp}}(\boldsymbol{\theta}, \mathbf{x}, y) = \mathcal{L}_{\text{ver}}(f(\boldsymbol{\theta}, \mathbf{x}), y)$. As outlined in §4.3, $\log \mathcal{L}_{\alpha,\text{Exp}}(\boldsymbol{\theta}, \mathbf{x}, y) = (1-\alpha)\log \mathcal{L}(f(\boldsymbol{\theta}, \mathbf{x}_{\text{adv}}), y) + \alpha\log \mathcal{L}_{\text{ver}}(f(\boldsymbol{\theta}, \mathbf{x}), y)$, which is continuous and monotonically increasing with respect to $\alpha$, and satisfies the following inequality for $\alpha \in [0,1]$:

$$\log \mathcal{L}(f(\boldsymbol{\theta}, \mathbf{x}_{\text{adv}}), y) \leq (1-\alpha)\log \mathcal{L}(f(\boldsymbol{\theta}, \mathbf{x}_{\text{adv}}), y) + \alpha\log \mathcal{L}_{\text{ver}}(f(\boldsymbol{\theta}, \mathbf{x}), y) \leq \log \mathcal{L}_{\text{ver}}(f(\boldsymbol{\theta}, \mathbf{x}), y).$$

By applying the exponential function, which, being monotonically increasing and continuous, preserves the monotonicity and continuity of $\log \mathcal{L}_{\alpha,\text{Exp}}(\boldsymbol{\theta}, \mathbf{x}, y)$, we obtain

$$\mathcal{L}(f(\boldsymbol{\theta}, \mathbf{x}_{\text{adv}}), y) \leq \mathcal{L}_{\alpha,\text{Exp}}(\boldsymbol{\theta}, \mathbf{x}, y) \leq \mathcal{L}_{\text{ver}}(f(\boldsymbol{\theta}, \mathbf{x}), y)\ \forall\,\alpha \in [0,1],$$

proving expressivity.

In order to prove that $\mathcal{L}_{\alpha,\text{Exp}}(\boldsymbol{\theta}, \mathbf{x}, y) \leq \mathcal{L}_{\alpha,\text{MTL}}(\boldsymbol{\theta}, \mathbf{x}, y)\ \forall\,\alpha \in [0,1]$, we start from:

$$\log\left((1-\alpha)\mathcal{L}(f(\boldsymbol{\theta}, \mathbf{x}_{\text{adv}}), y) + \alpha\,\mathcal{L}_{\text{ver}}(f(\boldsymbol{\theta}, \mathbf{x}), y)\right) \geq$$
$$(1-\alpha)\log \mathcal{L}(f(\boldsymbol{\theta}, \mathbf{x}_{\text{adv}}), y) + \alpha\log \mathcal{L}_{\text{ver}}(f(\boldsymbol{\theta}, \mathbf{x}), y),$$

which holds for any $\alpha \in [0,1]$ owing to the concavity of $\log$. By exponentiating both sides and recalling their definitions, we get $\mathcal{L}_{\alpha,\text{Exp}}(\boldsymbol{\theta}, \mathbf{x}, y) \leq \mathcal{L}_{\alpha,\text{MTL}}(\boldsymbol{\theta}, \mathbf{x}, y)\ \forall\,\alpha \in [0,1]$, concluding the proof. $\qquad\square$

### B.4 SABR

**Lemma B.1.** *For networks with monotonically increasing and continuous activation functions, if $\mathcal{C}(\mathbf{x}, \epsilon) := \{\mathbf{x}' : \|\mathbf{x}' - \mathbf{x}\|_\infty \leq \epsilon\}$ and when computed via IBP, $\underline{\mathbf{z}}_\lambda(\boldsymbol{\theta}, \mathbf{x}, y)$ as defined in equation (6) is continuous with respect to $\lambda$.*

*Proof.* For $\mathcal{C}(\mathbf{x}, \epsilon) := \{\mathbf{x}' : \|\mathbf{x}' - \mathbf{x}\|_\infty \leq \epsilon\}$, we can write $\mathcal{C}(\mathbf{x}, \epsilon) = [\mathbf{x} - \epsilon, \mathbf{x} + \epsilon] = [\mathbf{l}_0, \mathbf{u}_0]$ and $\mathcal{C}(\mathbf{x}_\lambda, \lambda\epsilon) = [\mathbf{l}_{0,\lambda}, \mathbf{u}_{0,\lambda}]$, with $\mathbf{l}_{0,\lambda} := \min(\max(\mathbf{x}_{\text{adv}}, \mathbf{l}_0 + \lambda\epsilon), \mathbf{u}_0 - \lambda\epsilon) - \lambda\epsilon$ and $\mathbf{u}_{0,\lambda} := \min(\max(\mathbf{x}_{\text{adv}}, \mathbf{l}_0 + \lambda\epsilon), \mathbf{u}_0 - \lambda\epsilon) + \lambda\epsilon$, which are continuous functions of $\lambda$ as $\max$, $\min$, linear combinations and compositions of continuous functions preserve continuity. The SABR bounds $\underline{\mathbf{z}}_\lambda(\boldsymbol{\theta}, \mathbf{x}, y)$ can then be computed as $\hat{\mathbf{l}}_n$ from equation (10), using $[\mathbf{l}_{0,\lambda}, \mathbf{u}_{0,\lambda}]$ as input domain. In this case, $\hat{\mathbf{u}}_1 = [W_k]_+\,\mathbf{u}_{0,\lambda} + [W_k]_-\mathbf{l}_{0,\lambda} + \mathbf{b}_k$ and $\hat{\mathbf{l}}_1 = [W_k]_+\,\mathbf{l}_{0,\lambda} + [W_k]_-\mathbf{u}_{0,\lambda} + \mathbf{b}_k$. The continuity of $\underline{\mathbf{z}}_\lambda(\boldsymbol{\theta}, \mathbf{x}, y)$ with respect to $\lambda$ then follows from its computation as the composition and linear combination of continuous functions. $\qquad\square$

**Proposition B.2.** *If $\mathcal{L}(\cdot, y)$ is continuous with respect to its first argument and $\underline{\mathbf{z}}_\lambda(\boldsymbol{\theta}, \mathbf{x}, y)$ as defined in equation (6) is continuous with respect to $\lambda \in [0,1]$, the SABR (Müller et al., 2023) loss function $\mathcal{L}(-\underline{\mathbf{z}}_\lambda(\boldsymbol{\theta}, \mathbf{x}, y), y)$ is expressive according to definition 3.1.*

*Proof.* Recalling equation (6), $\underline{\mathbf{z}}_\lambda(\boldsymbol{\theta}, \mathbf{x}, y) \leq \mathbf{z}(\boldsymbol{\theta}, \mathbf{x}', y)\ \forall\,\mathbf{x}' \in \mathcal{C}(\mathbf{x}_\lambda, \lambda\epsilon) \subseteq \mathcal{C}(\mathbf{x}, \epsilon)$, with $\mathbf{x}_\lambda := \text{Proj}(\mathbf{x}_{\text{adv}}, \mathcal{C}(\mathbf{x}, \epsilon - \lambda\epsilon))$. We can hence define $\delta_\lambda(\boldsymbol{\theta}, \mathbf{x}', y) := \underline{\mathbf{z}}_\lambda(\boldsymbol{\theta}, \mathbf{x}, y) - \mathbf{z}(\boldsymbol{\theta}, \mathbf{x}', y) \leq 0$ for any $\mathbf{x}' \in \mathcal{C}(\mathbf{x}_\lambda, \lambda\epsilon)$. As $\mathcal{C}(\mathbf{x}_{\lambda'}, \lambda'\epsilon) \subseteq \mathcal{C}(\mathbf{x}_\lambda, \lambda\epsilon)$ for any $\lambda' \leq \lambda$, $\underline{\mathbf{z}}_\lambda(\boldsymbol{\theta}, \mathbf{x}, y)$ will monotonically decrease with $\lambda$, and so will $\delta_\lambda(\boldsymbol{\theta}, \mathbf{x}', y)$. We can hence re-write the SABR loss as:

$$\mathcal{L}(-\underline{\mathbf{z}}_\lambda(\boldsymbol{\theta}, \mathbf{x}, y), y) = \mathcal{L}(-\left[\mathbf{z}(\boldsymbol{\theta}, \mathbf{x}_{\text{adv}}, y) + (\underline{\mathbf{z}}_\lambda(\boldsymbol{\theta}, \mathbf{x}, y) - \mathbf{z}(\boldsymbol{\theta}, \mathbf{x}_{\text{adv}}, y))\right], y)$$
$$= \mathcal{L}(-\left[\mathbf{z}(\boldsymbol{\theta}, \mathbf{x}_{\text{adv}}, y) + \delta_\lambda(\boldsymbol{\theta}, \mathbf{x}_{\text{adv}}, y)\right], y).$$

Then, $\delta_0(\boldsymbol{\theta}, \mathbf{x}_{\text{adv}}, y) = 0$, as $\mathbf{x}_0 = \mathbf{x}_{\text{adv}}$ and hence $\mathbf{z}_0(\boldsymbol{\theta}, \mathbf{x}, y) = \mathbf{z}(\boldsymbol{\theta}, \mathbf{x}_{\text{adv}}, y)$. Furthermore, $\underline{\mathbf{z}}_1(\boldsymbol{\theta}, \mathbf{x}, y) = \underline{\mathbf{z}}(\boldsymbol{\theta}, \mathbf{x}, y)$, proving the last two points of definition 3.1. Continuity follows from the fact that compositions of continuous functions ($\mathcal{L}(\cdot, y)$ and $\underline{\mathbf{z}}_\lambda(\boldsymbol{\theta}, \mathbf{x}, y)$) are continuous. The remaining requirements follow from using assumption 2.1 and the non-positivity and monotonicity of $\delta_\lambda(\boldsymbol{\theta}, \mathbf{x}_{\text{adv}}, y)$. $\qquad\square$

## C  LOSS DETAILS FOR CROSS-ENTROPY

Assuming the underlying loss function $\mathcal{L}$ to be cross-entropy, we now present an analytical form of the expressive losses presented in §4.1, §4.2, and §4.3, along with a comparison with the loss from SABR (Müller et al., 2023).

When applied on the logit differences $\mathbf{z}(\boldsymbol{\theta}, \mathbf{x}, y)$ (see §2.3), the cross-entropy can be written as $\mathcal{L}(-\mathbf{z}(\boldsymbol{\theta}, \mathbf{x}, y), y) = \log(1 + \sum_{i \neq y} e^{-\mathbf{z}(\boldsymbol{\theta}, \mathbf{x}, y)[i]})$ by exploiting the properties of logarithms and $\mathbf{z}(\boldsymbol{\theta}, \mathbf{x}, y)[y] = 0$ (see §2). In fact:

$$\mathcal{L}(-\mathbf{z}(\boldsymbol{\theta}, \mathbf{x}, y), y) = -\log\left(\frac{e^{-\mathbf{z}(\boldsymbol{\theta}, \mathbf{x}, y)[y]}}{\sum_i e^{-\mathbf{z}(\boldsymbol{\theta}, \mathbf{x}, y)[i]}}\right)$$

$$= -\log\left(\frac{1}{1 + \sum_{i \neq y} e^{-\mathbf{z}(\boldsymbol{\theta}, \mathbf{x}, y)[i]}}\right) = \log\left(1 + \sum_{i \neq y} e^{-\mathbf{z}(\boldsymbol{\theta}, \mathbf{x}, y)[i]}\right).$$

### C.1  CC-IBP

**Proposition C.1.** *When* $\mathcal{L}(-\mathbf{z}(\boldsymbol{\theta}, \mathbf{x}, y), y) = \log(1 + \sum_{i \neq y} e^{-\mathbf{z}(\boldsymbol{\theta}, \mathbf{x}, y)[i]})$, *the loss* $\mathcal{L}_{\alpha, CC}(\boldsymbol{\theta}, \mathbf{x}, y)$ *from equation* (7) *takes the following form:*

$$\mathcal{L}_{\alpha, CC}(\boldsymbol{\theta}, \mathbf{x}, y) = \log\left(1 + \sum_{i \neq y} \left(e^{-\mathbf{z}(\boldsymbol{\theta}, \mathbf{x}_{adv}, y)[i]}\right)^{(1-\alpha)} \left(e^{-\underline{\mathbf{z}}(\boldsymbol{\theta}, \mathbf{x}, y)[i]}\right)^{\alpha}\right).$$

*Proof.* Using the properties of exponentials:

$$\mathcal{L}_{\alpha, CC}(\boldsymbol{\theta}, \mathbf{x}, y) = \mathcal{L}(-\left[(1-\alpha)\,\mathbf{z}(\boldsymbol{\theta}, \mathbf{x}_{\text{adv}}, y) + \alpha\,\underline{\mathbf{z}}(\boldsymbol{\theta}, \mathbf{x}, y)\right], y)$$

$$= \log\left(1 + \sum_{i \neq y} e^{-[(1-\alpha)\,\mathbf{z}(\boldsymbol{\theta}, \mathbf{x}_{\text{adv}}, y) + \alpha\,\underline{\mathbf{z}}(\boldsymbol{\theta}, \mathbf{x}, y)][i]}\right)$$

$$= \log\left(1 + \sum_{i \neq y} \left(e^{-(1-\alpha)\,\mathbf{z}(\boldsymbol{\theta}, \mathbf{x}_{\text{adv}}, y)[i]}\right)\left(e^{-\alpha\,\underline{\mathbf{z}}(\boldsymbol{\theta}, \mathbf{x}, y)[i]}\right)\right)$$

$$= \log\left(1 + \sum_{i \neq y} \left(e^{-\mathbf{z}(\boldsymbol{\theta}, \mathbf{x}_{\text{adv}}, y)[i]}\right)^{(1-\alpha)}\left(e^{-\underline{\mathbf{z}}(\boldsymbol{\theta}, \mathbf{x}, y)[i]}\right)^{\alpha}\right).$$

$\qquad\square$

### C.2  MTL-IBP

**Proposition C.2.** *When* $\mathcal{L}(-\mathbf{z}(\boldsymbol{\theta}, \mathbf{x}, y), y) = \log(1 + \sum_{i \neq y} e^{-\mathbf{z}(\boldsymbol{\theta}, \mathbf{x}, y)[i]})$, *the loss* $\mathcal{L}_{\alpha, MTL}(\boldsymbol{\theta}, \mathbf{x}, y)$ *from equation* (8) *takes the following form:*

$$\mathcal{L}_{\alpha, MTL}(\boldsymbol{\theta}, \mathbf{x}, y) = \log\left(\left(1 + \sum_{i \neq y} e^{-\mathbf{z}(\boldsymbol{\theta}, \mathbf{x}_{adv}, y)[i]}\right)^{(1-\alpha)}\left(1 + \sum_{i \neq y} e^{-\underline{\mathbf{z}}(\boldsymbol{\theta}, \mathbf{x}, y)[i]}\right)^{\alpha}\right).$$

*Proof.* Using the properties of logarithms and exponentials:

$$\mathcal{L}_{\alpha,\text{MTL}}(\boldsymbol{\theta}, \mathbf{x}, y) = (1-\alpha)\mathcal{L}(f(\boldsymbol{\theta}, \mathbf{x}_{\text{adv}}), y) + \alpha\,\mathcal{L}_{\text{ver}}(f(\boldsymbol{\theta}, \mathbf{x}), y)$$

$$= (1-\alpha)\log\left(1 + \sum_{i \neq y} e^{-\mathbf{z}(\boldsymbol{\theta}, \mathbf{x}_{\text{adv}}, y)[i]}\right) + \alpha\log\left(1 + \sum_{i \neq y} e^{-\mathbf{z}(\boldsymbol{\theta}, \mathbf{x}, y)[i]}\right)$$

$$= \log\left(\left(1 + \sum_{i \neq y} e^{-\mathbf{z}(\boldsymbol{\theta}, \mathbf{x}_{\text{adv}}, y)[i]}\right)^{(1-\alpha)}\right) + \log\left(\left(1 + \sum_{i \neq y} e^{-\mathbf{z}(\boldsymbol{\theta}, \mathbf{x}, y)[i]}\right)^{\alpha}\right)$$

$$= \log\left(\left(1 + \sum_{i \neq y} e^{-\mathbf{z}(\boldsymbol{\theta}, \mathbf{x}_{\text{adv}}, y)[i]}\right)^{(1-\alpha)}\left(1 + \sum_{i \neq y} e^{-\mathbf{z}(\boldsymbol{\theta}, \mathbf{x}, y)[i]}\right)^{\alpha}\right).$$

$\square$

By comparing propositions C.1 and C.2, we can see that verifiability tends to play a relatively larger role on $\mathcal{L}_{\alpha,\text{MTL}}(\boldsymbol{\theta}, \mathbf{x}, y)$. In particular, given that $\mathbf{z}(\boldsymbol{\theta}, \mathbf{x}_{\text{adv}}, y) > \underline{\mathbf{z}}(\boldsymbol{\theta}, \mathbf{x}, y)$, if the network displays a large margin of empirical robustness to a given incorrect class ($\mathbf{z}(\boldsymbol{\theta}, \mathbf{x}_{\text{adv}}, y)[i] >> 0$, with $i \neq y$), the weight of the verifiability of that class on $\mathcal{L}_{\alpha,\text{CC}}(\boldsymbol{\theta}, \mathbf{x}, y)$ will be greatly reduced. On the other hand, if the network has a large margin of empirical robustness to all incorrect classes ($\mathbf{z}(\boldsymbol{\theta}, \mathbf{x}_{\text{adv}}, y) >> \mathbf{0}$), verifiability will still have a relatively large weight on $\mathcal{L}_{\alpha,\text{MTL}}(\boldsymbol{\theta}, \mathbf{x}, y)$. These observations comply with proposition 4.2, which shows that $\mathcal{L}_{\alpha,\text{CC}}(\boldsymbol{\theta}, \mathbf{x}, y) \leq \mathcal{L}_{\alpha,\text{MTL}}(\boldsymbol{\theta}, \mathbf{x}, y)$, indicating that $\mathcal{L}_{\alpha,\text{MTL}}(\boldsymbol{\theta}, \mathbf{x}, y)$ is generally more skewed towards verifiability.

### C.3    EXP-IBP

When $\mathcal{L}(-\mathbf{z}(\boldsymbol{\theta}, \mathbf{x}, y), y) = \log(1 + \sum_{i \neq y} e^{-\mathbf{z}(\boldsymbol{\theta}, \mathbf{x}, y)[i]})$, the loss $\mathcal{L}_{\alpha,\text{Exp}}(\boldsymbol{\theta}, \mathbf{x}, y)$ from equation (9) takes the following form:

$$\mathcal{L}_{\alpha,\text{Exp}}(\boldsymbol{\theta}, \mathbf{x}, y) = \log^{(1-\alpha)}\left(1 + \sum_{i \neq y}\left(e^{-\mathbf{z}(\boldsymbol{\theta}, \mathbf{x}_{\text{adv}}, y)[i]}\right)\right)\log^{\alpha}\left(1 + \sum_{i \neq y}\left(e^{-\mathbf{z}(\boldsymbol{\theta}, \mathbf{x}, y)[i]}\right)\right).$$

### C.4    COMPARISON OF CC-IBP WITH SABR

**Proposition C.3.** *When* $\mathcal{L}(-\mathbf{z}(\boldsymbol{\theta}, \mathbf{x}, y), y) = \log(1 + \sum_{i \neq y} e^{-\mathbf{z}(\boldsymbol{\theta}, \mathbf{x}, y)[i]})$, *the loss* $\mathcal{L}(-\underline{\mathbf{z}}_{\lambda}(\boldsymbol{\theta}, \mathbf{x}, y), y)$ *from equation* (6) *takes the following form:*

$$\mathcal{L}(-\underline{\mathbf{z}}_{\lambda}(\boldsymbol{\theta}, \mathbf{x}, y), y) = \log\left(1 + \sum_{i \neq y}\left(e^{-\mathbf{z}(\boldsymbol{\theta}, \mathbf{x}_{adv}, y)[i]}\right)\left(e^{-\delta_{\lambda}(\boldsymbol{\theta}, \mathbf{x}_{adv}, y)[i]}\right)\right),$$

*where* $\delta_{\lambda}(\boldsymbol{\theta}, \mathbf{x}_{adv}, y) \leq 0$ *is monotonically decreasing with* $\lambda$.

*Proof.* We can use $\delta_{\lambda}(\boldsymbol{\theta}, \mathbf{x}', y) := \underline{\mathbf{z}}_{\lambda}(\boldsymbol{\theta}, \mathbf{x}, y) - \mathbf{z}(\boldsymbol{\theta}, \mathbf{x}', y) \leq 0$, which is monotonically decreasing with $\lambda$ as seen in the proof of proposition B.2. Then, exploiting the properties of the exponential, we can trivially re-write the SABR loss $\mathcal{L}(-\underline{\mathbf{z}}_{\lambda}(\boldsymbol{\theta}, \mathbf{x}, y), y)$ as:

$$\mathcal{L}(-\underline{\mathbf{z}}_{\lambda}(\boldsymbol{\theta}, \mathbf{x}, y), y) = \mathcal{L}(-\left[\mathbf{z}(\boldsymbol{\theta}, \mathbf{x}_{\text{adv}}, y) + \delta_{\lambda}(\boldsymbol{\theta}, \mathbf{x}, y)\right], y)$$

$$= \log\left(1 + \sum_{i \neq y} e^{-\left[\mathbf{z}(\boldsymbol{\theta}, \mathbf{x}_{\text{adv}}, y) + \delta_{\lambda}(\boldsymbol{\theta}, \mathbf{x}, y)\right][i]}\right)$$

$$= \log\left(1 + \sum_{i \neq y}\left(e^{-\mathbf{z}(\boldsymbol{\theta}, \mathbf{x}_{\text{adv}}, y)[i]}\right)\left(e^{-\delta_{\lambda}(\boldsymbol{\theta}, \mathbf{x}, y)[i]}\right)\right).$$

$\square$

By comparing propositions C.1 and C.3, we can see that the SABR and CC-IBP losses share a similar form, and they both upper-bound the adversarial loss. The main differences are the following: (i) the adversarial term is scaled down in CC-IBP, (ii) CC-IBP computes lower bounds over the entire input region, (iii) the over-approximation term of CC-IBP uses a scaled lower bound of the logit differences, rather than its difference with the adversarial $\mathbf{z}$.

## D  PSEUDO-CODE

Algorithm 1 provides pseudo-code for CC-MTL, MTL-IBP and Exp-IBP.

---

**Algorithm 1** CC/MTL-IBP training loss computation

---

1: **Input:** Network $f(\boldsymbol{\theta}, \cdot)$, input $\mathbf{x}$, label $y$, perturbation set $\mathcal{C}(\mathbf{x}, \epsilon)$, loss function $\mathcal{L}$, hyper-parameters $\alpha, \lambda$
2: Compute $\mathbf{x}_{\mathrm{adv}}$ using an attack on $\mathcal{C}(\mathbf{x}, \epsilon)$
3: Compute $\underline{\mathbf{z}}(\boldsymbol{\theta}, \mathbf{x}, y)$ via IBP on $\mathcal{C}(\mathbf{x}, \epsilon)$ as described in section 2.2.1
4: **if** CC-IBP **then**
5:     $\mathbf{z}(\boldsymbol{\theta}, \mathbf{x}_{\mathrm{adv}}, y) \leftarrow (f(\boldsymbol{\theta}, \mathbf{x}_{\mathrm{adv}})[y]\,\mathbf{1} - f(\boldsymbol{\theta}, \mathbf{x}_{\mathrm{adv}}))$
6:     $\mathcal{L}_{\alpha}(\boldsymbol{\theta}, \mathbf{x}, y) \leftarrow \mathcal{L}(- \left[(1 - \alpha)\,\mathbf{z}(\boldsymbol{\theta}, \mathbf{x}_{\mathrm{adv}}, y) + \alpha\,\underline{\mathbf{z}}(\boldsymbol{\theta}, \mathbf{x}, y)\right], y)$
7: **else**
8:     $\mathcal{L}_{\mathrm{ver}}(f(\boldsymbol{\theta}, \mathbf{x}), y) \leftarrow \mathcal{L}(-\underline{\mathbf{z}}(\boldsymbol{\theta}, \mathbf{x}, y), y)$
9:     **if** MTL-IBP **then**
10:         $\mathcal{L}_{\alpha}(\boldsymbol{\theta}, \mathbf{x}, y) \leftarrow (1 - \alpha)\mathcal{L}(f(\boldsymbol{\theta}, \mathbf{x}_{\mathrm{adv}}), y) + \alpha\,\mathcal{L}_{\mathrm{ver}}(f(\boldsymbol{\theta}, \mathbf{x}), y)$
11:     **else if** Exp-IBP **then**
12:         $\mathcal{L}_{\alpha}(\boldsymbol{\theta}, \mathbf{x}, y) \leftarrow \mathcal{L}(f(\boldsymbol{\theta}, \mathbf{x}_{\mathrm{adv}}), y)^{(1 - \alpha)}\,\mathcal{L}_{\mathrm{ver}}(f(\boldsymbol{\theta}, \mathbf{x}), y)^{\alpha}$
13:     **end if**
14: **end if**
15: **return** $\mathcal{L}_{\alpha}(\boldsymbol{\theta}, \mathbf{x}, y) + \lambda||\boldsymbol{\theta}||_1$

---

## E  BRANCH-AND-BOUND FRAMEWORK

We now outline the details of the Branch-and-Bound (BaB) framework employed to verify the networks trained within this work (see §6).

Before resorting to more expensive verifiers, we first run IBP (Gowal et al., 2018) and CROWN (Zhang et al., 2018) from the automatic LiRPA library (Xu et al., 2020) as an attempt to quickly verify the property. If the attempt fails, we run BaB with a timeout of 1800 seconds per data point unless specified otherwise, possibly terminating well before the timeout (except on the MNIST experiments) when the property is deemed likely to time out. Similarly to a recent verified training work (De Palma et al., 2022), we base our verifier on the publicly available OVAL framework (Bunel et al., 2020a;b; De Palma et al., 2021c) from VNN-COMP-2021 (Bak et al., 2021).

**Lower bounds to logit differences**   A crucial component of a BaB framework is the strategy employed to compute $\underline{\mathbf{z}}(\boldsymbol{\theta}, \mathbf{x}, y)$. As explained in §2.2, the computation of the bound relies on a network over-approximation, which will generally depend on "intermediate bounds": bounds to intermediate network pre-activations. For ReLU network, the over-approximation quality will depend on the tightness of the bounds for "ambiguous" ReLUs, which cannot be replaced by a linear function (0 is contained in their pre-activation bounds). We first use IBP to inexpensively compute intermediate bounds for the whole network. Then, we refine the bounds associated to ambiguous ReLUs by using 5 iterations of $\alpha$-CROWN (Xu et al., 2021) with a step size of 1 decayed by 0.98 at every iteration, without jointly optimizing over them so as to reduce the memory footprint. Intermediate bounds are computed only once at the root node, that is, before any branching is performed. If the property is not verified at the root, we use $\beta$-CROWN (Wang et al., 2021) as bounding algorithm for the output bounds, with a dynamic number of iterations (see (De Palma et al., 2021c, section 5.1.2)), and initialized from the parent BaB sub-problem. We employ a step size of 0.1, decayed by 0.98 at each iteration.

**Adversarial attacks** Counter-examples to the properties are searched for by evaluating the network on the input points that concretize the lower bound $\mathbf{z}(\boldsymbol{\theta}, \mathbf{x}, y)$ according to the employed network over-approximation. This is repeated on each BaB sub-problem, and also used to mark sub-problem infeasibility when $\mathbf{z}(\boldsymbol{\theta}, \mathbf{x}, y)$ overcomes the corresponding input point evaluation. In addition, some of these points are used as initialization (along with random points in the perturbation region) for a 500-step MI-FGSM Dong et al. (2018) adversarial attack, which is run repeatedly for each property using a variety of randomly sampled hyper-parameter (step size value and decay, momentum) settings, and early stopped if success is deemed unlikely. In §6.2 we report a network to be empirically robust if both the above procedure and a 40-step PGD attack (Madry et al., 2018) (run before BaB) fail to find an adversarial example.

**Branching** Branching is performed through activation spitting: we create branch-and-bound sub-problems by splitting a ReLU into its two linear functions. In other words, given intermediate bounds on pre-activations $\hat{\mathbf{x}}_k \in [\hat{\mathbf{l}}_k, \hat{\mathbf{u}}_k]$ for $k \in [\![2, n-1]\!]$, if the split is performed at the $i$-th neuron of the $k$-th layer, $\hat{\mathbf{l}}_k[i] \geq 0$ is enforced on a sub-problem, $\hat{\mathbf{u}}_k[i] \leq 0$ on the other. Let us denote the Hadamard product by $\odot$. We employ a slightly modified version of the UPB branching strategy (De Palma et al., 2022, equation (7)), where the bias of the upper constraint from the Planet relaxation (ReLU convex hull, also known as triangle relaxation), $\frac{[-\hat{\mathbf{l}}_k]_+ \odot [\hat{\mathbf{u}}_k]_+}{\hat{\mathbf{u}}_k - \hat{\mathbf{l}}_k}$, is replaced by the area of the entire relaxation, $[-\hat{\mathbf{l}}_k]_+ \odot [\hat{\mathbf{u}}_k]_+$. Intuitively, the area of the over-approximation should better represent the weight of an ambiguous neuron in the overall neural network relaxation than the distance of the upper constraint from the origin. In practice, we found the impact on BaB performance to be negligible. In particular, we tested our variant against the original UPB, leaving the remaining BaB setup unchanged, on the CC-IBP trained network for MNIST with $\epsilon = 0.1$ from table 1: they both resulted in $98.43\%$ verified robust accuracy.

# F    EXPERIMENTAL DETAILS

We present additional details pertaining to the experiments from §6 and appendix G. In particular, we outline dataset and training details, hyper-parameters, computational setup and the employed network architecture.

## F.1    DATASETS

MNIST (LeCun et al., 2010) is a 10-class (each class being a handwritten digit) classification dataset consisting of $28 \times 28$ greyscale images, with 60,000 training and 10,000 test points. CIFAR-10 (Krizhevsky & Hinton, 2009) consists of $32 \times 32$ RGB images in 10 different classes (associated to subjects such as airplanes, cars, etc.), with 50,000 training and 10,000 test points. TinyImageNet (Le & Yang, 2015) is a miniature version of the ImageNet dataset, restricted to 200 classes of $64 \times 64$ RGB images, with 100,000 training and 10,000 validation images. The employed downscaled ImageNet (Chrabaszcz et al., 2017) is a 1000-class dataset of $64 \times 64$ RGB images, with 1,281,167 training and 50,000 validation images. Similarly to previous work (Xu et al., 2020; Shi et al., 2021; Müller et al., 2023), we normalize all datasets and use random horizontal flips and random cropping as data augmentation on CIFAR-10, TinyImageNet and downscaled ImageNet. Complying with common experimental practice Gowal et al. (2018); Xu et al. (2020); Shi et al. (2021); Müller et al. (2023), we train on the entire training sets, and evaluate MNIST and CIFAR-10 on their test sets, ImageNet and TinyImageNet on their validation sets.

## F.2    NETWORK ARCHITECTURE, TRAINING SCHEDULE AND HYPER-PARAMETERS

We outline the practical details of our training procedure.

**Network Architecture** Independently of the dataset, all our experiments employ a 7-layer architecture that enjoys widespread usage (with minor modifications) in the verified training literature (Gowal et al., 2018; Zhang et al., 2020; Shi et al., 2021; Müller et al., 2023; Mao et al., 2023). In particular, similarly to recent work (Müller et al., 2023; Mao et al., 2023), we employ a version modified by Shi et al. (2021) to use BatchNorm (Ioffe & Szegedy, 2015) after every layer. Differently

from Shi et al. (2021); Müller et al. (2023), at training time we compute IBP bounds to BatchNorm layers using the batch statistics from the perturbed data $\mathbf{x}_{\text{adv}}$. These are the same batch statistics employed to compute the training-time adversarial logit differences and their associated loss, which are not part of the loss functions used by previous work under the same architecture (Shi et al., 2021; Müller et al., 2023). As done by Müller et al. (2023), the training-time attacks are carried out with the network in evaluation mode. Finally, the statistics used at evaluation time are computed including both the original data and the attacks used during training. The details of the architecture, often named `CNN7` in the literature, are presented in table 2.

Table 2: Details of the `CNN7` architecture.

| `CNN7` |
| --- |
| Convolutional: 64 filters of size $3 \times 3$, stride 1, padding 1 |
| BatchNorm layer |
| ReLU layer |
| Convolutional layer: 64 filters of size $3 \times 3$, stride 1, padding 1 |
| BatchNorm layer |
| ReLU layer |
| Convolutional layer: 128 filters of size $3 \times 3$, stride 2, padding 1 |
| BatchNorm layer |
| ReLU layer |
| Convolutional layer: 128 filters of size $3 \times 3$, stride 1, padding 1 |
| BatchNorm layer |
| ReLU layer |
| Convolutional layer: 128 filters of size $3 \times 3$, stride 1 or $2^*$, padding 1 |
| BatchNorm layer |
| ReLU layer |
| Linear layer: 512 hidden units |
| BatchNorm layer |
| ReLU layer |
| Linear layer: as many hidden units as the number of classes |

$^*$1 for MNIST and CIFAR-10, 2 for TinyImageNet and ImageNet64.

**Train-Time Perturbation Radius** We train using the target perturbation radius (denoted by $\epsilon$ as elsewhere in this work) for all benchmarks except on MNIST and CIFAR-10 when $\epsilon = {}^2/_{255}$. On MNIST, complying with Shi et al. (2021), we use $\epsilon_{\text{train}} = 0.2$ for $\epsilon = 0.1$ and $\epsilon_{\text{train}} = 0.4$ for $\epsilon = 0.3$. On CIFAR-10 with $\epsilon = {}^2/_{255}$, in order to improve performance, SABR (Müller et al., 2023) "shrinks" the magnitude of the computed IBP bounds by multiplying intermediate bounds by a constant $c = 0.8$ before each ReLU (for instance, following our notation in §2.2 $[\hat{\mathbf{u}}_{k-1}]_+$ would be replaced by $0.8[\hat{\mathbf{u}}_{k-1}]_+$). As a result, the prominence of the attack in the employed loss function is increased. In other to achieve a similar effect, we employ the original perturbation radius to compute the lower bounds $\underline{\mathbf{z}}(\boldsymbol{\theta}, \mathbf{x}, y)$ and use a larger perturbation radius to compute $\mathbf{x}_{\text{adv}}$ for our experiments in this setup (see table 3). Specifically, we use $\epsilon_{\text{train}} = 4.2$, the radius employed by IBP-R (De Palma et al., 2022) to perform the attack and to compute the algorithm's regularization terms.

**Training Schedule, Initialization, Regularization** Following Shi et al. (2021), training proceeds as follows: the network is initialized according to the method presented by Shi et al. (2021), which relies on a low-variance Gaussian distribution to prevent the "explosion" of IBP bounds across layers at initialization. Training starts with the standard cross-entropy loss during warm-up. Then, the perturbation radius is gradually increased to the target value $\epsilon_{\text{train}}$ (using the same schedule

Table 3: Hyper-parameter configurations for the networks trained via CC-IBP, MTL-IBP and Exp-IBP from tables 1 and 4.

| Dataset | $\epsilon$ | Method | $\alpha$ | $\ell_1$ | Attack (steps, $\eta$) | $\epsilon_{\text{train}}$, if $\neq \epsilon$ | Total epochs | Warm-up + Ramp-up | LR decay epochs |
| --- | --- | --- | --- | --- | --- | --- | --- | --- | --- |
| MNIST | 0.1 | CC-IBP | $1 \times 10^{-1}$ | $2 \times 10^{-6}$ | $(1, 10.0\epsilon)$ | 0.2 | 70 | 0+20 | 50,60 |
| | | MTL-IBP | $1 \times 10^{-2}$ | $1 \times 10^{-5}$ | $(1, 10.0\epsilon)$ | 0.2 | 70 | 0+20 | 50,60 |
| | 0.3 | CC-IBP | $3 \times 10^{-1}$ | $3 \times 10^{-6}$ | $(1, 10.0\epsilon)$ | 0.4 | 70 | 0+20 | 50,60 |
| | | MTL-IBP | $8 \times 10^{-2}$ | $1 \times 10^{-6}$ | $(1, 10.0\epsilon)$ | 0.4 | 70 | 0+20 | 50,60 |
| CIFAR-10 | $\frac{2}{255}$ | CC-IBP | $1 \times 10^{-2}$ | $3 \times 10^{-6}$ | $(8, 0.25\epsilon)$ | $^{4.2}/_{255}$ attack only | 160 | 1+80 | 120,140 |
| | | MTL-IBP | $4 \times 10^{-3}$ | $3 \times 10^{-6}$ | $(8, 0.25\epsilon)$ | $^{4.2}/_{255}$ attack only | 160 | 1+80 | 120,140 |
| | | Exp-IBP | $9.5 \times 10^{-2}$ | $4 \times 10^{-6}$ | $(8, 0.25\epsilon)$ | $^{4.2}/_{255}$ attack only | 160 | 1+80 | 120,140 |
| | $\frac{8}{255}$ | CC-IBP | $5 \times 10^{-1}$ | 0 | $(1, 10.0\epsilon)$ | / | 260 | 1+80 | 180,220 |
| | | MTL-IBP | $5 \times 10^{-1}$ | $1 \times 10^{-7}$ | $(1, 10.0\epsilon)$ | / | 260 | 1+80 | 180,220 |
| | | Exp-IBP | $5 \times 10^{-1}$ | 0 | $(1, 10.0\epsilon)$ | / | 260 | 1+80 | 180,220 |
| TinyImageNet | $\frac{1}{255}$ | CC-IBP | $1 \times 10^{-2}$ | $5 \times 10^{-5}$ | $(1, 10.0\epsilon)$ | / | 160 | 1+80 | 120,140 |
| | | MTL-IBP | $1 \times 10^{-2}$ | $5 \times 10^{-5}$ | $(1, 10.0\epsilon)$ | / | 160 | 1+80 | 120,140 |
| | | Exp-IBP | $4 \times 10^{-2}$ | $5 \times 10^{-5}$ | $(1, 10.0\epsilon)$ | / | 160 | 1+80 | 120,140 |
| ImageNet64 | $\frac{1}{255}$ | CC-IBP | $5 \times 10^{-2}$ | $1 \times 10^{-5}$ | $(1, 10.0\epsilon)$ | / | 80 | 1+20 | 60,70 |
| | | MTL-IBP | $5 \times 10^{-2}$ | $1 \times 10^{-5}$ | $(1, 10.0\epsilon)$ | / | 80 | 1+20 | 60,70 |
| | | Exp-IBP | $5 \times 10^{-2}$ | $1 \times 10^{-5}$ | $(1, 10.0\epsilon)$ | / | 80 | 1+20 | 60,70 |

as Shi et al. (2021)) for a fixed number of epochs during the ramp-up phase. During warm-up and ramp-up, the regularization introduced by Shi et al. (2021) is added to the objective. This is composed of two terms: one that penalizes (similarly to the initialization) bounds explosion at training time, the other that balances the impact of active and inactive ReLU neurons in each layer. Finally, towards the end of training, and after ramp-up has ended, the learning rate is decayed twice. In line with previous work (Xiao et al., 2019; De Palma et al., 2022; Müller et al., 2023; Mao et al., 2023), we employ $\ell_1$ regularization throughout training.

**Hyper-parameters**   The hyper-parameter configurations used in table 1 are presented in table 3. The training schedules for MNIST and CIFAR-10 with $\epsilon = {}^2/_{255}$ follow the hyper-parameters from Shi et al. (2021). Similarly to Müller et al. (2023), we increase the number of epochs after ramp-up for CIFAR-10 with $\epsilon = {}^8/_{255}$: we use 260 epochs in total. On TinyImageNet, we employ the same configuration as CIFAR-10 with $\epsilon = {}^2/_{255}$, and ImageNet64 relies on the TinyImageNet schedule from Shi et al. (2021), who did not benchmark on it. Following previous work (Shi et al., 2021; Müller et al., 2023), we use a batch size of 256 for MNIST, and of 128 everywhere else. As Shi et al. (2021); Müller et al. (2023), we use a learning rate of $5 \times 10^{-5}$ for all experiments, decayed twice by a factor of 0.2, and use gradient clipping with threshold equal to 10. Coefficients for the regularization introduced by Shi et al. (2021) are set to the values suggested in the author's official codebase: $\lambda = 0.5$ on MNIST and CIFAR-10, $\lambda = 0.2$ for TinyImageNet. On ImageNet64, which was not included in their experiments, we use $\lambda = 0.2$ as for TinyImageNet given the dataset similarities. Single-step attacks were forced to lay on the boundary of the perturbation region (see appendix G.9), while we complied with previous work for the 8-step PGD attack (Balunovic & Vechev, 2020; De Palma et al., 2022). As Shi et al. (2021), we clip target epsilon values to the fifth decimal digit. The main object of our tuning were the $\alpha$ coefficients from the CC-IBP, MTL-IBP and Exp-IBP losses (see §4) and the $\ell_1$ regularization coefficient: for both, we first selected an order of magnitude, then the significand via manual search. We found Exp-IBP to be significantly more sensitive to the $\alpha$ parameter compared to CC-IBP and MTL-IBP on some of the benchmarks. We do not perform early stopping. Unless otherwise stated, all the ablation studies re-use the hyper-parameter configurations from table 3. In particular, the ablation from §6.2 use the single-step attack for CIFAR-10 with $\epsilon = {}^2/_{255}$. Furthermore, the ablation from appendix G.4 uses $\alpha = 5 \times 10^{-2}$ for CC-IBP and $\alpha = 1 \times 10^{-2}$ for MTL-IBP in order to ease verification.

### F.3   COMPUTATIONAL SETUP

While each experiment (both for training and post-training verification) was run on a single GPU at a time, we used a total of 6 different machines and 12 GPUs. In particular, we relied on the following CPU models: Intel i7-6850K, i9-7900X CPU, i9-10920X, i9-10940X, AMD 3970X. The employed GPU models for most of the experiments were: Nvidia Titan V, Titan XP, and Titan XP. The experiments of §6.3 and appendix G.5 on CIFAR-10 for $\epsilon = {}^8/_{255}$ were run on the following GPU models: RTX 4070 Ti, RTX 2080 Ti, RTX 3090. 8 GPUs of the following models: Nvidia Titan V, Nvidia Titan XP, and Nvidia Titan XP.

### F.4   SOFTWARE ACKNOWLEDGMENTS

Our training code is mostly built on the publicly available training pipeline by Shi et al. (2021), which was released under a 3-Clause BSD license. We further rely on the automatic LiRPA library (Xu et al., 2020), also released under a 3-Clause BSD license, and on the OVAL complete verification framework (Bunel et al., 2020a; De Palma et al., 2021c), released under a MIT license. MNIST and CIFAR-10 are taken from `torchvision.datasets` (Paszke et al., 2019); we downloaded TinyImageNet from the Stanford CS231n course website (`http://cs231n.stanford.edu/TinyImageNet-200.zip`), and downscaled ImageNet from the ImageNet website (`https://www.image-net.org/download.php`).

## G   ADDITIONAL EXPERIMENTAL RESULTS

We now present experimental results complementing those in §6.

Table 4: Comparison of CC-IBP and MTL-IBP with literature results for $\ell_\infty$ norm perturbations on the MNIST dataset. The entries corresponding to the best standard or verified robust accuracy for each perturbation radius are highlighted in bold. For the remaining entries, improvements on the literature performance of the best-performing ReLU-based architectures are underlined.

| $\epsilon$ | Method | Source | Standard acc. [%] | Verified rob. acc. [%] |
|---|---|---|---|---|
| | CC-IBP | this work | **99.30** | **98.43** |
| | MTL-IBP | this work | 99.25 | 98.38 |
| 0.1 | TAPS | Mao et al. (2023) | 99.19 | 98.39 |
| | SABR | Müller et al. (2023) | 99.23 | 98.22 |
| | SORTNET | Zhang et al. (2022b) | 99.01 | 98.14 |
| | IBP* | Mao et al. (2024) | 98.86 | 98.23 |
| | ADVIBP | Fan & Li (2021) | 98.97 | 97.72 |
| | CROWN-IBP | Zhang et al. (2020) | 98.83 | 97.76 |
| | COLT | Balunovic & Vechev (2020) | 99.2 | 97.1 |
| | CC-IBP | this work | **98.81** | 93.58 |
| | MTL-IBP | this work | 98.80 | 93.62 |
| 0.3 | STAPS | Mao et al. (2023) | 98.53 | 93.51 |
| | SABR* | Mao et al. (2024) | 98.48 | **93.85** |
| | SORTNET | Zhang et al. (2022b) | 98.46 | 93.40 |
| | IBP* | Mao et al. (2024) | 97.66 | 93.35 |
| | ADVIBP | Fan & Li (2021) | 98.42 | 91.77 |
| | CROWN-IBP | Zhang et al. (2020) | 98.18 | 92.98 |
| | COLT | Balunovic & Vechev (2020) | 97.3 | 85.7 |

*$4\times$ wider network than the architecture employed in our experiments.

## G.1 COMPARISON WITH LITERATURE RESULTS ON MNIST

We present a comparison of CC-IBP and MTL-IBP with results from the literature on the MNIST (Le-Cun et al., 2010) dataset. Table 4 shows that both methods attain state-of-the-art performance on both perturbation radii, further confirming the results from table 1 on the centrality of expressivity for verified training. Nevertheless, we point out that, owing to its inherent simplicity, performance differences on MNIST are smaller than those seen on the datasets employed in §6.1.

## G.2 TRAINING TIMES

Table 5 reports training times for CC-IBP and MTL-IBP, and present a comparison with the training times, whenever available in the source publications, associated to the literature results from tables 1 and 4. Our timings from the table were measured on a Nvidia Titan V GPU and using an 20-core Intel i9-7900X CPU. We point out that, as literature measurements were carried out on different hardware depending on the source, these only provide a rough indication of the overhead associated with each algorithm. Owing to the our use of a single-step attack on all benchmarks except one, MTL-IBP and CC-IBP tend to incur less training overhead than other state-of-the-art approaches, making them more scalable and easier to tune. In particular, STAPS displays significantly larger training overhead. As expected, under the same training schedule, the training times of CC-IBP and MTL-IBP are roughly equivalent. While the computational setup of our measurements from table 5 was no longer available at the time of the Exp-IBP experiments, the same equivalence applies. For instance, on an Nvidia GeForce RTX 4070 Ti and a 24-core Intel i9-10920X CPU, training with CC-IBP, MTL-IBP and Exp-IBP on TinyImageNet respectively takes $4.60 \times 10^4$, $4.57 \times 10^4$, and $4.59 \times 10^4$ seconds.

## G.3 ABLATION: POST-TRAINING VERIFICATION

Table 6 reports the verified robust accuracy of the models trained with CC-IBP, MTL-IBP and Exp-IBP from tables 1 and 4 under the verification schemes outlined in §6.2. As expected, the verified robust accuracy decreases when using inexpensive incomplete verifiers. The difference between BaB and CROWN/IBP accuracies varies with the setting, with larger perturbations displaying smaller gaps. Furthermore, CROWN bounds do not improve over those obtained via IBP on larger perturbations, mirroring previous observations on IBP-trained networks (Zhang et al., 2020). We note that all three methods already attain larger verified robustness than the literature results from table 1 under CROWN/IBP bounds on TinyImageNet and downscaled ImageNet.

Table 5: Training times of CC-IBP and MTL-IBP compared to literature results for $\ell_\infty$ norm perturbations on MNIST, CIFAR-10, TinyImageNet and downscaled ($64 \times 64$) ImageNet datasets.

| Dataset | $\epsilon$ | Method | Source | Training time [s] |
|---|---|---|---|---|
| MNIST | 0.1 | CC-IBP | this work | $2.71 \times 10^3$ |
| | | MTL-IBP | this work | $2.69 \times 10^3$ |
| | | TAPS | Mao et al. (2023) | $4.26 \times 10^4$ |
| | | SABR | Müller et al. (2023) | $1.22 \times 10^4$ |
| | | SORTNET | Zhang et al. (2022b) | $1.59 \times 10^4$ |
| | | ADVIBP | Fan & Li (2021) | $1.29 \times 10^5$ |
| | 0.3 | CC-IBP | this work | $2.71 \times 10^3$ |
| | | MTL-IBP | this work | $2.72 \times 10^3$ |
| | | STAPS | Mao et al. (2023) | $1.24 \times 10^4$ |
| | | SORTNET | Zhang et al. (2022b) | $1.59 \times 10^4$ |
| | | ADVIBP | Fan & Li (2021) | $1.29 \times 10^5$ |
| | | CROWN-IBP | Zhang et al. (2020) | $5.58 \times 10^3$ |
| CIFAR-10 | $\frac{2}{255}$ | CC-IBP | this work | $1.77 \times 10^4$ |
| | | MTL-IBP | this work | $1.76 \times 10^4$ |
| | | STAPS | Mao et al. (2023) | $1.41 \times 10^5$ |
| | | SORTNET | Zhang et al. (2022b) | $4.04 \times 10^4$ |
| | | ADVIBP | Fan & Li (2021) | $5.28 \times 10^5$ |
| | | CROWN-IBP | Zhang et al. (2020) | $9.13 \times 10^4$ |
| | | COLT | Balunovic & Vechev (2020) | $1.39 \times 10^5$ |
| | $\frac{8}{255}$ | CC-IBP | this work | $1.72 \times 10^4$ |
| | | MTL-IBP | this work | $1.70 \times 10^4$ |
| | | STAPS | Mao et al. (2023) | $2.70 \times 10^4$ |
| | | SABR | Müller et al. (2023) | $2.64 \times 10^4$ |
| | | SORTNET | Zhang et al. (2022b) | $4.04 \times 10^4$ |
| | | IBP-R | De Palma et al. (2022) | $5.89 \times 10^3$ |
| | | IBP | Shi et al. (2021) | $9.51 \times 10^3$ |
| | | ADVIBP | Fan & Li (2021) | $5.28 \times 10^5$ |
| | | CROWN-IBP | Xu et al. (2020) | $5.23 \times 10^4$ |
| | | COLT | Balunovic & Vechev (2020) | $3.48 \times 10^4$ |
| TinyImageNet | $\frac{1}{255}$ | CC-IBP | this work | $6.58 \times 10^4$ |
| | | MTL-IBP | this work | $6.56 \times 10^4$ |
| | | STAPS | Mao et al. (2023) | $3.06 \times 10^5$ |
| | | SORTNET | Zhang et al. (2022b) | $1.56 \times 10^5$ |
| | | IBP | Shi et al. (2021) | $3.53 \times 10^4$ |
| | | CROWN-IBP | Shi et al. (2021) | $3.67 \times 10^4$ |
| ImageNet64 | $\frac{1}{255}$ | CC-IBP | this work | $3.26 \times 10^5$ |
| | | MTL-IBP | this work | $3.52 \times 10^5$ |
| | | SORTNET | Zhang et al. (2022b) | $6.58 \times 10^5$ |

$^*4\times$ wider network than the architecture employed in our experiments and in the original work (Müller et al., 2023).

## G.4 ABLATION: ATTACK TYPES

Table 7 examines the effect of the employed attack type (how to compute $\mathbf{x}_{\text{adv}}$ in equations (7) and (8)) on the performance of CC-IBP and MTL-IBP, focusing on CIFAR-10. PGD denotes an 8-step PGD (Madry et al., 2018) attack with constant step size $\eta = 0.25\epsilon$, FGSM a single-step attack with $\eta \geq 2.0\epsilon$, whereas None indicates that no attack was employed (in other words, $\mathbf{x}_{\text{adv}} = \mathbf{x}$). While FGSM and PGD use the same hyper-parameter configuration for CC-IBP and MTL-IBP from table 1, $\alpha$ is raised when $\mathbf{x}_{\text{adv}} = \mathbf{x}$ in order to increase verifiability (see appendix F). As shown in table 7, the use of adversarial attacks, especially for $\epsilon = 2/255$, are crucial to attain state-of-the-art verified robust accuracy. Furthermore, the use of inexpensive attacks only adds a minimal overhead (roughly 15% on $\epsilon = 2/255$). If an attack is indeed employed, neither CC-IBP nor MTL-IBP appear to be particularly sensitive to the strength of the adversary for $\epsilon = 8/255$, where FGSM slightly outperforms PGD. On the other hand, PGD improves the overall performance on $\epsilon = 2/255$, leading both CC-IBP and MTL-IBP to better robustness-accuracy trade-offs than those reported by previous work on this benchmark. In general, given the overhead and the relatively small performance difference, FGSM appears to be preferable to PGD on most benchmarks.

Table 6: Verified robust accuracy under different post-training verification schemes for networks trained via CC-IBP and MTL-IBP from tables 1 and 4.

| Dataset | $\epsilon$ | Method | Verified robust accuracy [%] | | |
|---|---|---|---|---|---|
| | | | BaB | CROWN/IBP | IBP |
| MNIST | 0.1 | CC-IBP | 98.43 | 96.37 | 94.30 |
| | | MTL-IBP | 98.38 | 97.17 | 96.67 |
| | 0.3 | CC-IBP | 93.46 | 92.22 | 92.22 |
| | | MTL-IBP | 93.62 | 92.45 | 92.45 |
| CIFAR-10 | $\frac{2}{255}$ | CC-IBP | 63.78 | 52.37 | 0.00 |
| | | MTL-IBP | 63.24 | 51.35 | 0.00 |
| | | Exp-IBP | 61.65 | 47.87 | 0.00 |
| | $\frac{8}{255}$ | CC-IBP | 35.27 | 34.02 | 34.02 |
| | | MTL-IBP | 35.44 | 34.64 | 34.64 |
| | | Exp-IBP | 35.04 | 33.88 | 33.88 |
| TinyImageNet | $\frac{1}{255}$ | CC-IBP | 26.39 | 23.58 | 0.05 |
| | | MTL-IBP | 26.09 | 23.65 | 0.11 |
| | | Exp-IBP | 26.18 | 22.96 | 0.01 |
| ImageNet64 | $\frac{1}{255}$ | CC-IBP | 11.87 | 10.64 | 0.92 |
| | | MTL-IBP | 12.13 | 10.87 | 1.41 |
| | | Exp-IBP | 13.30 | 11.42 | 0.19 |

## G.5 TEST EVALUATION OF MODELS FROM §6.3

Table 8 reports the results obtained by training the $\alpha$ values for CC-IBP, MTL-IBP and SABR found in figure 2 on the entirety of the CIFAR-10 training set, and evaluating on the test set. All the other hyper-parameters comply with those detailed in appendix F. In order to reflect the use of $\epsilon_{\text{train}} = {}^{4.2}/_{255}$ for the attack computation in CC-IBP and MTL-IBP for $\epsilon = {}^2/_{255}$ (see appendix F), on that setup we compute the SABR loss over the train-time perturbation radius of $\epsilon_{\text{train}} = {}^{4.2}/_{255}$. Branch-and-bound is run with a timeout of 600 seconds. These three expressive losses display similar trade-offs between verified robustness and standard accuracy, with CC-IBP slightly outperforming the other algorithms on $\epsilon = {}^2/_{255}$ and MTL-IBP on $\epsilon = {}^8/_{255}$. Observe that SABR yields a strictly better trade-off than the one reported by the original authors (79.24% and 62.84% standard and verified robust accuracy, respectively (Müller et al., 2023)), highlighting the importance of careful tuning. Given that, for $\epsilon = {}^2/_{255}$, the "small box" may be well out of the target perturbation set, the result suggest that the performance of SABR itself is linked to definition 3.1 rather than to the selection of a subset of the target perturbation set containing the worst-case input. We remark that this may be in contrast with the explanation from the authors (Müller et al., 2023). By comparing with literature results from table 1 (typically directly tuned on the test set, see appendix F.2), and given the heterogeneity across the three losses considered in this experiment, we can conclude that satisfying expressivity leads to state-of-the-art results on ReLU networks. Finally, the test results confirm that minimizing the approximation of the branch-and-bound loss does not necessarily correspond to superior robustness-accuracy trade-offs (see §6.3).

Table 7: Effect of the type of adversarial attack on the performance of CC-IBP and MTL-IBP under $\ell_\infty$ perturbations on the CIFAR-10 test set.

| $\epsilon$ | Method | Attack | Accuracy [%] | | Train. time [s] |
|---|---|---|---|---|---|
| | | | Std. | Ver. rob. | |
| $\frac{2}{255}$ | CC-IBP | FGSM | 80.22 | 62.94 | $1.08 \times 10^4$ |
| | CC-IBP | None | 82.12 | 38.80 | $9.40 \times 10^3$ |
| | CC-IBP | PGD | 80.09 | 63.78 | $1.77 \times 10^4$ |
| | MTL-IBP | FGSM | 79.70 | 62.67 | $1.08 \times 10^4$ |
| | MTL-IBP | None | 81.22 | 36.09 | $9.33 \times 10^3$ |
| | MTL-IBP | PGD | 80.11 | 63.24 | $1.76 \times 10^4$ |
| $\frac{8}{255}$ | CC-IBP | FGSM | 53.71 | 35.27 | $1.72 \times 10^4$ |
| | CC-IBP | None | 57.95 | 33.21 | $1.48 \times 10^4$ |
| | CC-IBP | PGD | 53.12 | 35.17 | $2.82 \times 10^4$ |
| | MTL-IBP | FGSM | 53.35 | 35.44 | $1.70 \times 10^4$ |
| | MTL-IBP | None | 55.62 | 33.62 | $1.44 \times 10^4$ |
| | MTL-IBP | PGD | 53.30 | 35.19 | $2.83 \times 10^4$ |

Table 8: Performance of the CC-IBP, MTL-IBP and SABR models resulting from the validation study presented in figure 2 under $\ell_\infty$ norm perturbations on the CIFAR-10 test set. The results corresponding to the best trade-off across tuning methods (measured as the sum of standard and verified robust accuracy) is highlighted in bold.

| $\epsilon$ | Method | $\alpha$ | Tuning criterion | Accuracy [%] | | |
| --- | --- | --- | --- | --- | --- | --- |
| | | | | Std. | Ver. rob. | Std. + Ver. rob. |
| | CC-IBP | $1 \times 10^{-2}$ | BaB Loss + Std. Loss | 79.95 | 63.70 | **143.65** |
| | CC-IBP | $3 \times 10^{-3}$ | BaB Err | 81.93 | 61.12 | 143.05 |
| $\frac{2}{255}$ | MTL-IBP | $7 \times 10^{-3}$ | BaB Loss + Std. Loss | 79.26 | 63.33 | **142.59** |
| | MTL-IBP | $3 \times 10^{-3}$ | BaB Err | 79.86 | 62.43 | 142.29 |
| | SABR | $1 \times 10^{-2}$ | BaB Loss + Std. Loss | 79.69 | 63.33 | **143.02** |
| | SABR | $3 \times 10^{-3}$ | BaB Err | 81.48 | 60.01 | 141.49 |
| | CC-IBP | $5 \times 10^{-1}$ | BaB Loss + Std. Loss | 53.62 | 35.11 | **88.73** |
| | CC-IBP | $7 \times 10^{-1}$ | BaB Err | 51.45 | 35.29 | 86.74 |
| $\frac{8}{255}$ | MTL-IBP | $3 \times 10^{-1}$ | BaB Loss + Std. Loss | 57.15 | 33.84 | 90.99 |
| | MTL-IBP | $3 \times 10^{-1}$ | BaB Err | 57.15 | 33.84 | 90.99 |
| | SABR | $4 \times 10^{-1}$ | BaB Loss + Std. Loss | 57.23 | 33.47 | **90.70** |
| | SABR | $7 \times 10^{-1}$ | BaB Err | 51.95 | 35.01 | 86.96 |

## G.6 TUNING SABR ON TINYIMAGENET AND IMAGENET64

The results of the validation study on CIFAR-10 (table G.5) showed that SABR, which satisfies our definition of expressivity from §3, yields similar performance to CC-IBP and MTL-IBP when employing the same tuning methodology. Prompted by these results, we carried out our own tuning of SABR on TinyImageNet and downscaled ImageNet, where the gap between our expressive losses (CC-IBP, MTL-IBP and Exp-IBP) and the results from the literature is the largest (see §6.1). In order to reduce overhead and in line with the results from table 1 and previous work (Gowal et al., 2018; Zhang et al., 2020; Shi et al., 2021; Müller et al., 2023), we directly tuned on the evaluation sets. The resulting SABR hyper-parameters are identical to those employed for CC-IBP on each of the two datasets: see table 3. Table 9 shows that SABR displays similar performance profiles to our expressive losses from §4 (except for Exp-IBP on ImageNet64: see the relative remark in §6.1), confirming the link between satisfying definition 3.1 and attaining state-of-the-art verified training performance. We ascribe the differences with respect to previously reported SABR results Müller et al. (2023); Mao et al. (2024) on TinyImageNet to insufficient tuning and inadequate regularization: for instance, Müller et al. (2023) set $\alpha = 0.4$ and $\lambda = 10^{-6}$ ($\ell_1$ coefficient); we employ $\alpha = 10^{-2}$ and $\lambda = 5 \times 10^{-5}$.

## G.7 LOSS SENSITIVITY TO $\alpha$

Figure 4 compares the standard, adversarial (under a 40-step PGD attack with $\eta = 0.035\epsilon$) and verified losses (under IBP and CROWN/IBP bounds) of the networks from figure 1 with the CC-IBP, MTL-

Table 9: Comparison of the expressive losses from §4 with our own SABR (Müller et al., 2023) experiments for $\ell_\infty$ norm perturbations on TinyImageNet and downscaled ($64 \times 64$) ImageNet. The entries corresponding to the best standard or verified robust accuracy for each perturbation radius are highlighted in bold.

| Dataset | $\epsilon$ | Method | Source | Standard acc. [%] | Verified rob. acc. [%] |
| --- | --- | --- | --- | --- | --- |
| | | CC-IBP | this work | 38.61 | **26.39** |
| | | MTL-IBP | this work | 37.56 | 26.09 |
| | | EXP-IBP | this work | **38.71** | 26.18 |
| TinyImageNet | $\frac{1}{255}$ | SABR | this work | 38.68 | 25.85 |
| | | SABR | Müller et al. (2023) | 28.85 | 20.46 |
| | | SABR[†] | Mao et al. (2024) | 28.97 | 21.36 |
| | | CC-IBP | this work | 19.62 | 11.87 |
| | | MTL-IBP | this work | 20.15 | 12.13 |
| ImageNet64 | $\frac{1}{255}$ | EXP-IBP | this work | **22.73** | **13.30** |
| | | SABR | this work | 20.33 | 12.39 |

[†] $2\times$ wider network than the architecture used in our experiments and the original work Müller et al. (2023).

IBP and Exp-IBP losses (see §4) computed with the corresponding training hyper-parameters. In all cases, their behavior closely follows the adversarial loss, which they upper bound. By comparing with figure 1, we conclude that the losses perform at their best in terms of trade-offs between BaB accuracy and standard accuracy when the losses used for training lower bound losses resulting from inexpensive incomplete verifiers. All verified losses converge to similar values for increasing $\alpha$ coefficients.

### G.8 EXPERIMENTAL VARIABILITY

In order to provide an indication of experimental variability for the results from §6, we repeat a small subset of our CC-IBP and MTL-IBP experiments from table 1, those on CIFAR-10 with $\epsilon = 8/255$, a further three times. Table 10 reports the mean, maximal and minimal values in addition to the sample standard deviation for both the standard and verified robust accuracies. In all cases, the experimental variability is relatively small.

Table 10: Experimental variability for the performance of CC-IBP and MTL-IBP for $\ell_\infty$ norm perturbations of radius $\epsilon = 8/255$ on the CIFAR-10 test set.

| Method | Standard accuracy [%] | | | | Verified robust accuracy [%] | | | |
|---|---|---|---|---|---|---|---|---|
| | Mean | Std. | Max | Min | Mean | Std. | Max | Min |
| CC-IBP | 54.04 | 0.23 | 54.23 | 53.71 | 35.06 | 0.29 | 35.34 | 34.73 |
| MTL-IBP | 53.77 | 0.40 | 54.18 | 53.35 | 35.25 | 0.18 | 35.44 | 35.05 |

### G.9 ABLATION: ADVERSARIAL ATTACK STEP SIZE

Table 11: Effect of the attack step size $\eta$ on the performance of CC-IBP and MTL-IBP under $\ell_\infty$ norm perturbations on the CIFAR-10 test set.

| $\epsilon$ | Method | Attack $\eta$ | Accuracy [%] | |
|---|---|---|---|---|
| | | | Std. | Ver. rob. |
| $\frac{2}{255}$ | CC-IBP | 10.0 $\epsilon$ | 80.22 | 62.94 |
| | CC-IBP | 1.25 $\epsilon$ | 80.94 | 62.50 |
| | MTL-IBP | 10.0 $\epsilon$ | 79.70 | 62.67 |
| | MTL-IBP | 1.25 $\epsilon$ | 80.43 | 62.29 |
| $\frac{8}{255}$ | CC-IBP | 10.0 $\epsilon$ | 53.71 | 35.27 |
| | CC-IBP | 1.25 $\epsilon$ | 54.75 | 34.77 |
| | MTL-IBP | 10.0 $\epsilon$ | 53.35 | 35.44 |
| | MTL-IBP | 1.25 $\epsilon$ | 54.30 | 35.38 |

Section G.4 investigates the influence of the type of attack used to compute $\mathbf{z}(\boldsymbol{\theta}, \mathbf{x}_{\text{adv}}, y)$ at training time on the performance of CC-IBP and MTL-IBP. We now perform a complementary ablation by keeping the attack fixed to randomly-initialized FGSM (Goodfellow et al., 2015; Wong et al., 2020) and changing the attack step size $\eta$. The results from §6 (except for the PGD entry in table 7) employ a step size $\eta = 10.0 \, \epsilon$ (any value larger than $2.0 \, \epsilon$ yields the same effect) in order to force the attack on the boundary of the perturbation region. Specifically, we report the performance for $\eta = 1.25 \, \epsilon$,

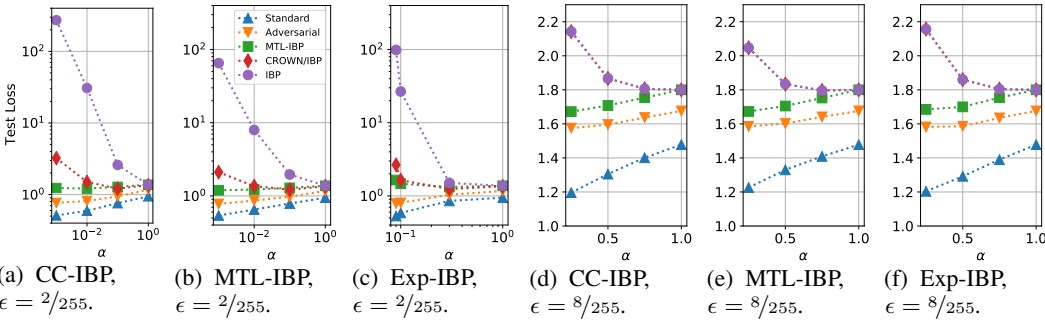

(a) CC-IBP, $\epsilon = 2/255$.  (b) MTL-IBP, $\epsilon = 2/255$.  (c) Exp-IBP, $\epsilon = 2/255$.  (d) CC-IBP, $\epsilon = 8/255$.  (e) MTL-IBP, $\epsilon = 8/255$.  (f) Exp-IBP, $\epsilon = 8/255$.

Figure 4: Loss values, computed on the full CIFAR-10 test set, for the models from Figure 1. The CC-IBP, MTL-IBP and Exp-IBP losses (see §4) are computed using the $\alpha$ value employed for training, denoted on the $x$ axis. The legend in plot 4(b) applies to all sub-figures.

which was shown by Wong et al. (2020) to prevent catastrophic overfitting. Table 11 does not display significant performance differences depending on the step size, highlighting the lack of catastrophic overfitting. We speculate that this is linked to the regularizing effect of the IBP bounds employed within CC-IBP and MTL-IBP. On all considered settings, the use of larger $\eta$ results in larger standard accuracy, and a decreased verified robust accuracy.

