# OpenReview forum: "Expressive Losses for Verified Robustness via Convex Combinations"
_ICLR.cc/2024/Conference — ICLR 2024 poster_

### Official Review · Reviewer_eo5p · 2023-10-19

**Soundness:** 3 good
**Presentation:** 3 good
**Contribution:** 1 poor
**Rating:** 8
**Confidence:** 4

**Summary:**

This paper proposes a scheme for convexly combining verified and adversarial loss functions to train verifiably robust models.

**Strengths:**

1. The paper is generally well written and presented.

2. The contribution is generally clear.

3. The idea of expressive loss functions as interpolating between verified and adversarial losses is interesting.

**Weaknesses:**

1. My main concern is that the paper is not sufficiently novel to merit publication. Convexly combining loss functions is a rather obvious idea; indeed (8) is just a linear combination of two loss functions, something which has been around for ages.

**Questions:**

1. In table 3, the alpha parameter for the MTL-IBP method can get very low (e.g., $4 \cdot 10^{-3}$ for CIFAR-10 $2/255$). Does this not mean that the loss essentially just reduces to the adversarial loss?

2. Why do the optimal $alpha$'s vary so much between the $2/255$ and $8/255$ epsilons for CIFAR-10? As in Q1, the MTL-IBP loss boils down to just the adversarial loss for $2/255$, and changes to a $50/50$ split for $8/255$.

---

> ### Author Response · Authors · 2023-11-20
> **Response to reviewer eo5p**
>
> We thank the reviewer for their time and feedback. We are glad that the reviewer found the paper to be well written and presented, and idea of expressive losses to be interesting.
>
> > My main concern is that the paper is not sufficiently novel to merit publication. Convexly combining loss functions is a rather obvious idea; indeed (8) is just a linear combination of two loss functions, something which has been around for ages.
>
> We are happy that the reviewer pointed out that convexly combining loss functions is a rather obvious idea that has been around for ages: we absolutely agree!
> Indeed, *one of the main contributions of our work is precisely to show that ideas that are as simple as they are old yield state-of-the-art performance when employed, for the first-time, to meet our novel definition of expressivity (Definition 3.1).* We believe that these results provide strong support for the centrality of the notion of expressivity for verified training, the core message of our submission, as well as a novel understanding of the recent literature.
> In other words, our aim is to show that the performance of recent successes in the literature can be matched by simple techniques, and explained through a very simple definition. We believe this to be at least as valuable for the community as introducing a novel and complex algorithm that outperforms the baselines on selected benchmarks.
>
> > In table 3, the alpha parameter for the MTL-IBP method can get very low  (e.g., $4 \cdot 10^{-3}$ for CIFAR-10 $2/255$). Does this not mean that the loss essentially just reduces to the adversarial loss?
>
> We thank the reviewer for the question. As visible in Figures 4a and 4b in appendix F.4, the adversarial and CC/MTL-IBP losses are fairly different even when $\alpha=10^{-3}$. Specifically (we report the raw numbers here for convenience), for the CC-IBP-trained network, $L_{adv}=0.7658$ and $L_{CC-IBP}=1.234$. For the one trained via MTL-IBP: $L_{adv}=0.777$ and $L_{MTL-IBP}=1.192$. This is due to the extremely large values that the IBP loss takes on under these low over-approximation coefficients: $L_{IBP}=272.159$ and $L_{IBP}=65.392$ for the networks trained via CC-IBP and MTL-IBP, respectively.
> As a result, even when $\alpha = 4 \cdot 10^{-3}$, the MTL-IBP loss still features a significant IBP component.
>
> > Why do the optimal alpha's vary so much between the $2/255$ and $8/255$ epsilons for CIFAR-10?
>
> We thank the reviewer for the interesting question. The difference between the type of loss that works the best on smaller perturbations and on larger perturbations on CIFAR-10 is a well-known phenomenon in the literature. Methods that have a strong adversarial training component (IBP-R, COLT) tend to perform particularly well on the $2/255$ setting, and quite poorly when $\epsilon=8/255$, where methods exclusively based on over-approximations typically outperform them (IBP, CROWN-IBP).
> Intuitively, larger perturbations will yield significantly larger bounds, requiring larger over-approximation coefficients $\alpha$ to effectively regularize them at the cost of a drop in standard accuracy.
>
>
> We sincerely hope the reviewer can reevaluate our contribution in light of the above responses, and remain available for further discussions and/or clarifications.

---

> > ### Comment · Reviewer_eo5p · 2023-11-21
> >
> > Thank you to the authors for their thorough clarifications and I apologize for the lack of detail in my original review. Having carefully read the other reviews, I will elaborate on my main concern with the paper.
> >
> > The authors define a notion of loss family expressiveness, which they claim is "key" for effective verified training (also noting that the SABR loss is expressive). To support their claim, the authors present two specific expressive loss families and demonstrate that these losses yield strong verified accuracy after searching over $\alpha$.
> >
> > To assess the experimental support provided in the paper, we must first understand rigorously what the authors mean by saying that expressive loss families are "key." Does this mean that expressive loss families are _necessary_ for good verification, _sufficient_ for good verification, or both? I don't think this paper provides good support for either necessariness or sufficiency.
> >
> > __Why I'm not convinced of necessariness of expressive loss families:__
> >
> > 1. While the convex combination loss functions presented by the authors achieve good performance, so do other loss functions in the experiments which don't match the expressive family structure (e.g., the alpha beta crown loss).
> >
> > 2. The paper simply provides no evidence as to why expressive loss families are needed. Providing examples of expressive loss functions that work well is not the same as showing that the expressive loss family structure is necessary. The kind of evidence that I would find convincing might look something like the following (completely hypothetical). Say you had a simple synthetic dataset which is parameterized by some $\alpha \in [0,1]$. This parameterization might be constructed such that for $\alpha=0$, the optimal classifier minimizes the adversarial loss, while for $\alpha=1$, the optimal classifier minimizes the verified loss. And for intermediate $\alpha$'s, the optimal classifier minimizes the corresponding interpolated loss from the expressive family. While this exact construction might not be possible, this is the flavor of evidence that would convince me that considering the entire expressive loss family $\alpha \in [0,1]$ is important.
> >
> > __Why I'm not convinced of the sufficiency of expressive loss families:__
> >
> > 1. To show sufficiency, I would need to see an explanation as to why nothing outside the expressive loss family structure would be beneficial. For example, we could also consider interpolating three different losses: the clean loss, the adversarial loss, and the verified loss. This structure would subsume expressive loss functions as a special case. You could of course debunk this particular construction experimentally by showing that interpolating three losses doesn't provide a benefit over interpolating just the adversarial and verified losses. But this doesn't disprove the possibility that there is some other loss family which is better suited than the presented expressive loss family.
> >
> > 2. Even if two specific expressive loss families work well, there could still be some other expressive loss family which performs poorly. A pessimistic reader could think that CC-IBP and MTL-IBP are just cherry picked examples of expressive loss families which support the conclusion of the paper. The paper doesn't convincingly argue that expressive loss families _must_ lead to good verification performance.
> >
> > __Summary__
> >
> > In my judgement, the paper doesn't provide convincing evidence that expressive loss families are either necessary or sufficient. Thus a pessimistic take on the paper might summarize it as "they cook up two new losses which work decently and coincidentally share this particular structure." I don't see enough in this paper to rebut this perspective. Please feel free to clarify if I am misunderstanding something.

---

> > > ### Author Response · Authors · 2023-11-21
> > > **Response reviewer eo5p's comment (1/2)**
> > >
> > > We sincerely thank the reviewer for taking the time to reply to our response, and for the detailed feedback. We will first reply to the main concern being raised, and then individually respond to the more specific comments.
> > >
> > > > we must first understand rigorously what the authors mean by saying that expressive loss families are "key." Does this mean that expressive loss families are necessary for good verification, sufficient for good verification, or both?
> > >
> > > We do not claim expressivity to be necessary to attain state-of-the-art performance, and we do not exclude other algorithms or frameworks could outperform the current state-of-the-art.
> > > We never formally claim sufficiency either: we believe formally demonstrating expressivity to be sufficient in a setting of practical relevance to be an exciting item for future work, if at all possible.
> > > Nevertheless, we provide a comprehensive body of evidence showing that expressivity is an empirically important concept for verified training, which is linked with strong performance and to recent advances in the literature, which it generalizes:
> > >
> > > - We show that arguably the simplest and most obvious ways to satisfy expressivity (convex combinations, as the reviewer stated in their original review, are `a rather obvious idea`), already reach state-of-the-art performance (Table 1), at least matching the performance of SABR (ICLR23) and (S)TAPS (NeurIPS23) on all considered benchmarks.
> > > - We show that for CC-IBP, MTL-IBP and SABR (three arguably rather different forms of expressive losses), maximising performance involves exploring a fine-grained interval between the adversarial and the verified loss (Figures 2 and 3), and that different parts of the $[0, 1]$ interval should be considered depending on the benchmark. Furthermore, better approximations of the worst-case loss do not necessarily result in better performance, contradicting a common view in previous work (SABR/TAPS) and strengthening the role of expressivity (spanning $[0,1]$) alone.
> > > - We show that CC-IBP, MTL-IBP and SABR attain arguably similar performance profiles when tuned on a holdout validation set according to the same criteria (Table 6).
> > >
> > > While our work is empirical in nature, we believe these to be worthy contributions of their own, and of interest to the verified training community (as confirmed by the comments from reviewer oQBq).

---

> > > > ### Author Response · Authors · 2023-11-21
> > > > **Response reviewer eo5p's comment (2/2)**
> > > >
> > > > > While the convex combination loss functions presented by the authors achieve good performance, so do other loss functions in the experiments which don't match the expressive family structure (e.g., the alpha beta crown loss).
> > > >
> > > > We are unsure what the reviewer means by the $\alpha$-$\beta$-CROWN loss: if this refers to the BaB loss in Figure 3, this is computed post-training on a small validation set, and never employed for training (this would be infeasible, as each batch point would require several seconds per logit difference). The models are trained using the method explained in the subcaptions (and the $\alpha$ values on the $x$ axis).
> > > > The only baseline methods that achieve good performance consistently through benchmarks are SABR and STAPS. While the former is expressive (as explained in section 3), the latter includes SABR as a special case, and does not consistently outperform it.
> > > >
> > > > > To show sufficiency, I would need to see an explanation as to why nothing outside the expressive loss family structure would be beneficial.  For example, we could also consider interpolating three different losses: the clean loss, the adversarial loss, and the verified loss.  [...] this doesn't disprove the possibility that there is some other loss family which is better suited than the presented expressive loss family.
> > > >
> > > > We do not exclude that other loss families or frameworks may outperform expressive losses. Nevertheless, as highlighted by the results in Table 1, no existing method currently outperforms them: further improving performance is an exciting direction for future work. Concerning the reviewer's suggestion: given that the adversarial loss upper bounds the standard loss and that pure adversarial training notoriously yields low certified accuracy, we do not believe that including the standard loss in the interpolation could be beneficial (see also the ablation in Table 5 of appendix F.2).
> > > >
> > > > > Even if two specific expressive loss families work well, there could still be some other expressive loss family which performs poorly. A pessimistic reader could think that CC-IBP and MTL-IBP are just cherry picked examples of expressive loss families which support the conclusion of the paper. The paper doesn't convincingly argue that expressive loss families must lead to good verification performance.
> > > >
> > > > As explained above, we would argue that CC-IBP, MTL-IBP and SABR are diverse enough to support our conclusion that expressivity is a key concept for state-of-the-art verified training. We disagree that proving necessity or sufficiency would be necessary for our work to be of interest (and impact) for the community. Nevertheless, we would be happy to test any expressive loss for which the reviewer believes a worse performance should be expected.
> > > >
> > > > > a pessimistic take on the paper might summarize it as "they cook up two new losses which work decently and coincidentally share this particular structure." I don't see enough in this paper to rebut this perspective.
> > > >
> > > > Given the consistently good performance across benchmarks, and notably the relatively large margin with which the results from the literature are outperformed on TinyImageNet and ImageNet64, we respectfully disagree that "decent" is an appropriate wording for the performance of CC-IBP and MTL-IBP. They both attain state-of-the-art performance, in spite of their striking simplicity. Exactly because of their simplicity and because of their shared properties with SABR, we disagree that our work can be reduced to the reviewer's sentence. However, we would also argue that introducing novel state-of-the-art algorithms (which we do, as a by-product) would be a sufficiently novel contribution of its own.

---

> ### Comment · Reviewer_eo5p · 2023-11-21
>
> Thank you to the authors for their thorough rebuttal. I'm not completely convinced, but I will raise my score to a five for now.
>
> "Nevertheless, we would be happy to test any expressive loss for which the reviewer believes a worse performance should be expected."
>
> If the authors can present an additional, qualitatively distinct example of an expressive loss family which achieves good performance I think that would strengthen the paper considerably and I would find that convincing. I can't immediately think of more ways to interpolate the adversarial and verified loss.
>
> Since the deadline is tomorrow, just testing on one dataset would be acceptable (say, CIFAR10 8/255).

---

> > ### Author Response · Authors · 2023-11-22
> > **New loss in response to reviewer eo5p**
> >
> > We are deeply grateful to the reviewer for their timely and very constructive feedback.
> >
> > > If the authors can present an additional, qualitatively distinct example of an expressive loss family which achieves good performance I think that would strengthen the paper considerably and I would find that convincing. [...] just testing on one dataset would be acceptable (say, CIFAR10 8/255).
> >
> > We sincerely thank the reviewer for the suggestion.
> > We experimented with $$L_{exp} = L_{adv} ^ {1 - \alpha}  L_{ver} ^ {\alpha},$$ which we provisionally call Exp-IBP when using the IBP verified loss, on CIFAR $8/255$.
> > Expressivity can be easily shown by exploiting the properties of monotonically increasing functions and using $\log(L_{exp}) = (1 - \alpha) \log(L_{adv}) +  \alpha \log(L_{ver})$.
> > In order to reduce runtime, we set a timeout of $300$ seconds (as opposed to $1800$ in Table 1) when computing the verified robust accuracy.
> > As shown in the table below, Exp-IBP, with $\alpha=0.4$ (the remaining hyper-parameters as CC-IBP in Table 3) performs on par with CC-IBP and MTL-IBP, hence providing further evidence on the crucial role of expressivity for verified training.
> >
> > | Method    | Source | Std. Acc. | Ver. Rob. Acc. |
> > | -------- | ------- | -------- | ------- |
> > | SABR  | Mueller et al. (2023)  |  52.38 | 35.13 |
> > | STAPS  | Mao et al. (2023a)  |  52.82 | 34.65 |
> > | CC-IBP  | Table 1 (this work)  |  53.71 | 35.27 |
> > | MTL-IBP  | Table 1 (this work)  |  53.35 | 35.44 |
> > | **Exp-IBP**  | new experiment (this work)  |  **56.34** | **34.55** |
> >
> >
> > We thank again the reviewer for their constructive feedback, which will strengthen our work. We hope that their evaluation can take the new experiment into account.

---

> > > ### Comment · Reviewer_eo5p · 2023-11-22
> > >
> > > Thank you to the authors for their timely additional experiments. I think this work would be a good contribution to ICLR and am recommending acceptance. I would suggest that the authors include Exp-IBP in the final paper (even if it’s just in the appendix). It helps demonstrate the diversity of expressive loss families and strengthens the experimental claims of the paper.

---

> > > > ### Author Response · Authors · 2023-11-22
> > > > **Thank you for your positive reevaluation**
> > > >
> > > > We sincerely thank reviewer eo5p for the constructive, professional and timely exchange that led to their positive reevaluation of our submission. We will definitely include the presented Exp-IBP results in the final paper, and thank them again for leading us to execute the experiment, which will indeed strengthen our experimental claims.

---

### Official Review · Reviewer_oQBq · 2023-10-31

**Soundness:** 3 good
**Presentation:** 3 good
**Contribution:** 3 good
**Rating:** 8
**Confidence:** 4

**Summary:**

The authors hypothesize that expressive loss functions yield better training for verified robustness. A family of loss functions $\mathcal{L}_\alpha$ is expressive if
- $\mathcal{L}(f(\theta, x_\text{adv}), y) \leq \mathcal{L}_\alpha(\theta, x, y) \leq \mathcal{L}_v (f(\theta,x),y)$ for all $\alpha \in [0,1]$, where the left/right inequality become an equality if $\alpha = 0/1$ respectively.
- $\mathcal{L}_\alpha$ is monotonically increasing with $\alpha$

They support their hypothesis that expressive loss functions yield better training for verified robustness by showing that trivial expressive losses obtained via convex combinations between adversarial attacks and IBP bounds yield sota results. They further hypothesize that the notion of expressivity is crucial to get sota and argue as follows:
- They state that sota verified training algorithms rely on coupling adversarial attacks with over-approximations. They show that SABR is expressive - hence the good performance of SABR is inline with their explanation.
- Other expressive losses can be trivially designed via convex combinations, i.e.
	1. CC-IBP: combine adversarial and over-approximated network outputs with in the loss
	2. MTL-IBP: combine adversarial and verified losses
- Experimental evaluation of CC-IBP and MTL-IBP. Both attain sota, particularly on TinyImageNet and downscaled ImageNet.
Further, they analyze the parameter $\alpha$ governing a robustness-accuracy trade-off. Better approximations of the worst case loss do not necessarily correspond to performance improvements.

The authors experimentally compare CC-IBP and MTL-IBP to prior work and find that the proposed methods match or outperform the literature. They also study the effect of the over-approximation coefficient on the performance profiles of expressive losses. The take away here is that better approximations of the branch-and-bound loss do not necessarily result in better performance.

Observed that standard accuracy decreases with alpha and verified accuracy increases with alpha. The adversarial and verified robust accuracies unter tighter verifiers may first increase and then decrease with $\alpha$, hence the need for careful tuning according to the desired robustness-accuracy trade-off.

Finally, the assumption that better approximations of the worst-case loss results in better trade-offs between verified robustness and accuracy is investigated. The parameter $\alpha$ is chosen based on the performance on a hold out set consisting of 20% of the training set. The worst case loss is approximated using a branch-and-bound loss. The authors report that sometimes it is better for the BaB error to be positive and sometimes for the BaB error to be negative.

**Strengths:**

- The presentation is mostly good.
- Training certifiable networks is a relevant research problem.
- The ideas are conceptually simple yet seem to be effective.
- The work unifies and generalizes successfull approaches.
- The authors provided code.

**Weaknesses:**

- It remains unclear how stable the results are (for example w.r.t. different seeds).
- Writing could be improved in some parts of the paper, i.e. Section 6.3.
- It remains unclear what "tricks" i.e. for regularization and initialization where specifically used.

**Questions:**

- What are the confidence intervals for the results in the paper with respect to different seeds? Are the trends consistent w.r.t. the randomness due to different seeds?
- What specialized initialization and regularization techniques where used?

---

> ### Author Response · Authors · 2023-11-20
> **Response to reviewer oQBq**
>
> We thank reviewer oQBq for their comprehensive review, summary, and positive evaluation of the contributions we presented. We are particularly happy that the reviewer recognized that our submission "unifies and generalizes" previous work, and that the ideas we present are effective despite being "conceptually simple". We indeed believe these to be amongst the most important contributions in our work.
>
> > unclear how stable the results are (for example w.r.t. different seeds) [...] Are the trends consistent w.r.t. the randomness due to different seeds?
>
> We provide an indication of experimental variability in Table 7 in appendix F.5, which reports results under four total runs on CIFAR-10 under perturbations of radius $\epsilon=8/255$. Both MTL-IBP and CC-IBP display relatively small experimental variability in the experiment.
>
> > It remains unclear what "tricks" i.e. for regularization and initialization where specifically used. [...] What specialized initialization and regularization techniques where used?
>
> Similarly to relevant previous work (SABR, (S)TAPS), we employ the initialization and regularization techniques introduced by (Shi et al., 2021), in addition to $\ell_1$ regularization.
> More specifically, (Shi et al. 2021) propose to initialize weights with a low-variance Gaussian distribution to prevent the "explosion" of IBP bounds across layers at initialisation. Furthermore, during the warm-up and ramp-up phases of training (see appendix E.2, "Training Schedule"), (Shi et al. 2021) add two regularization terms to the objective: one that penalizes (similarly to the initialization) bounds explosion at training time, the other that balances the impact of active and inactive ReLU neurons in each layer.
> We will be happy to include this discussion and further explanations in the next revision of the paper, and thank the reviewer for their question.

---

### Official Review · Reviewer_2HZC · 2023-11-07

**Soundness:** 3 good
**Presentation:** 1 poor
**Contribution:** 2 fair
**Rating:** 6
**Confidence:** 2

**Summary:**

The paper proposes "expressive" loss functions that interpolate between IBP and adversarial loss in a simple linear combinations and show good empirical performance on a variety of datasets.

**Strengths:**

The empirical results seem strong, especially considering the fact that the proposed methods are simple interpolations.

**Weaknesses:**

The presentation and writing needs a lot of work and it seems the paper is hurriedly written. Specific concerns are below.


	The mathematical definition of property P in Eq 1 is given in section 2 without any discussion of what it means or entails and why is it interesting/useful.

	You could add atleast one example of how x_adv could possibly be generated in the background section.

	Explicitly write down what “verification” means before using it in section 2.1. I don’t know what the following statement means: “However, formal verification methods fail to formally prove their robustness in a feasible time”

	 “As seen from Eq 1, network is provably robust if …logit differences … all positive” – why and how before even defining what verification is.

	“Incomplete verifiers will only prove a subset of the properties” – which properties ? only one is defined.

	The unlaballed equation with relationship of \underline{z} and z. I would not write \underline{z} as an inequation when saying it a lower bound without defining it first say using an example.

	Can you give an example when o and l are not equal ? the definition of z requires o to be atleast as big as l, and the definition of z also makes sense only if o and l are equal. Is o the size of the output before softmax evaluation or after? I am having a hard time reconciling dimensions of f( ) and z( ) for translation-invariance.

	Are the other methods also grid-searched for best hyperparameters  for their respective methods ?

	The runtimes are a bit misleading? Does the runtime also include hyperameter search cost including the cost for best interpolating parameter ?

**Questions:**

Please see the weakness section.

---

> ### Author Response · Authors · 2023-11-20
> **Response to reviewer 2HZC**
>
> We thank reviewer 2HZC for their detailed feedback on the manuscript, which will improve the clarity of the writing. We will make sure that the next revisions will clarify the raised concerns about the presentation, which we address individually below.
> We are glad that the reviewer found our empirical results to be strong particularly given the simplicity of the losses we employ. Indeed, we believe that our results point to the centrality of expressivity to state-of-the-art training, providing a new understanding of the recent literature.
>
> > The mathematical definition of property P in Eq 1 [...]
>
> Equation (1) provides a standard definition of adversarial robustness: no point in a neighborhood of the inputs should be misclassified. This is expressed by saying that all the differences to the ground truth should be positive.
> The use of argmin instead of min was a typo, and we thank the reviewer for indirectly pointing us to this.
>
> > add atleast one example of how $x_{adv}$ could possibly be generated in the background section.
>
> We will provide an example in the appendix in the next revision. As we describe in section 2.1, $x_{adv}$ is obtained by running an adversarial attack, the most common example being PGD by Madry et al. (2018). In short, for $\ell_\infty$ perturbations, PGD proceeds by taking steps in the direction of the sign of the gradient of the network with respect to the input point, clipping to the allowed input perturbation set after every iteration.
>
> > Explicitly write down what “verification” means before using it in section 2.1. I don’t know what the following statement means: “However, formal verification methods fail to formally prove their robustness in a feasible time”
>
> The fact that a network is robust to a specific adversarial attack does not imply that it will be robust to stronger/unseen attacks. Verification methods provide guarantees that no attack will ever succeed and, in the worst-case, incur an exponential runtime (in the number of neurons), which is in practice infeasible for even medium-sized networks. Verified training algorithms allow to verify larger networks in reasonable time, leading to larger verified robust accuracies.
>
> > “As seen from Eq 1, network is provably robust if …logit differences … all positive” why and how before even defining what verification is.
>
> The sentence in question does not depend on the notion of formal verification. If one can show that, for $\forall x_0 \in \mathcal{C}(x, \epsilon)$, all logit differences with incorrect classes are positive, then no adversarial example (misclassified point in $\mathcal{C}(x, \epsilon)$) can exist.
>
> > “Incomplete verifiers will only prove a subset of the properties” – which properties ? only one is defined.
>
> Each robustness property $P(f(\theta, x), y)$ is a function of the point $x$ and its label $y$. As a result, a given dataset will have as many robustness properties as the number of input/label pairs. We will clarify this in the text in the next revision. Given that, as outlined in section 2.2.1, incomplete verifiers produce lower bounds on (4), a positive bound implies that the solution to (4) is positive too, providing a solution to the verification problem. A negative lower bound, instead, leaves the property at hand undecided.
>
> > Can you give an example when o and l are not equal ? [...]  Is o the size of the output before softmax evaluation or after?
>
> For multi-class classification, the label is a scalar ($l=1$), as explained before equation (1), and the output dimension of the network ($o$) is equal to the number of classes. As $z$ is the difference between a broadcast scalar (the entry of the network output corresponding to the ground truth) and the vector output of the network, $z$ and $f$ have the same dimensions.
>
> > Are the other methods also grid-searched
>
> As common in related work (see for instance SABR, (S)TAPS) the baseline results in Table 1 are taken directly from the literature, and refer to the best results ever reported for any given method in previous work (for instance, across any network architecture). In other words, each baseline was tuned by the authors of the works referenced in the "Source" entry of Table 1.
>
> >  The runtimes are a bit misleading? Does the runtime also include hyperameter search cost including the cost for best interpolating parameter ?
>
> Runtimes for any algorithm (either ours or a baseline) only refer to a single training run. We would like to point out that the baselines with the best average performance across benchmarks (SABR, (S)TAPS) all feature at least as many hyper-parameters as CC-IBP and MTL-IBP.
>
> We hope to have provided more clarity on the reviewer's concerns, and would be happy to provide further clarifications as needed.

---

### Official Review · Reviewer_tVkU · 2023-11-10

**Soundness:** 3 good
**Presentation:** 3 good
**Contribution:** 2 fair
**Rating:** 5
**Confidence:** 2

**Summary:**

This work studies the certified training with over-approximation for robustness certification. Specifically, the authors introduce the idea of expressivity of loss functions and show that it can range from worst-case loss to verified loss, based on which two forms of loss are proposed. The experiments show the performance of the new losses and some findings regarding robustness and accuracy are given.

**Strengths:**

- The motivation of the paper is sound, and the underlying theory regarding certified training remains unknown and challenging.
- The paper is generally well-organized and easy to follow.
- The experiments are comprehensive and different datasets and attack radii are used for the evaluation.

**Weaknesses:**

- My biggest concern is that the contribution and novelty of the paper are incremental and minor, which is about the expressivity of losses. However, it seems that it somehow borrows the idea of the previous work SABR, which gives an effective loss ranging from adversarial loss and verified loss.  The difference between this work and SABR is not that clear and significant as SABR can induce expressivity by letting $\lambda=\alpha$ as shown in Sec. 3.
- Some key details are not given in the main text. E.g., it is not clear from the main text how the logit differences are associated with an adversarial attack for CC-IBP when it is compared to CROWN-IBP in Sec. 4.1, without which the contribution and novelty are further weakened in terms of the comparison.
- The insight and intuition of the relationship between CC-IBP and MTL-IBP are not clear. For example, does any case exist where one can be degraded to the other? If so, either theoretical or empirical results are needed to show it.
- For Table 1, the proposed method is with BaB as a complete method, I wonder if it is fair to compare with some incomplete baselines.

**Questions:**

See the Weakness part.

---

> ### Author Response · Authors · 2023-11-20
> **Response to reviewer tVkU**
>
> We thank reviewer tVkU for their time, questions and comments.
> Before addressing them individually, we would like to highlight that, as explained in the general response, the main goal and contribution of our work is the idea that *expressivity is all you need for state-of-the-art verified training*. We show that extremely simple expressive losses (CC-IBP and MTL-IBP), obtained by casting, for the first time, standard techniques (convex combinations) under the lens of our novel definition, already yield state-of-the-art performance.
> We are glad that the reviewer found the paper to be well-organized and with a comprehensive evaluation, which we believe should be seen in support of the above point.
>
> > the contribution and novelty of the paper are incremental and minor, which is about the expressivity of losses. However, it seems that it somehow borrows the idea of the previous work SABR [...]
>
> As highlighted by reviewer oQBq, our aim is to unify and generalize previous approaches, providing a novel understanding of the state-of-the-art. Therefore, the fact that SABR satisfies our definition of expressivity (as we show in section 3) is intentional, and exactly part of what we aim to show: expressivity is the key to recent advances in the certified training literature.
>
> > it is not clear from the main text how the logit differences are associated with an adversarial attack for CC-IBP when it is compared to CROWN-IBP in Sec. 4.1, without which the contribution and novelty are further weakened in terms of the comparison.
>
> By "logit differences associated with an adversarial attack", we mean the vector of differences between the ground truth logit ($y$ denotes the ground truth class) and the other logits: $z(\theta, x_{\text{adv}}, y) := f(\theta, x_{\text{adv}})[y] - f(\theta,x_{\text{adv}})$ (see definition in section 2, and the definition of the CC-IBP loss in equation (7)). As described in section 2.1, $x_{\text{adv}}$ is obtained by running an adversarial attack on the network. In our experiments, as described in the second paragraph of section 6, $x_{\text{adv}}$ is output by a single-step attack (random-initialized FGSM) in all settings except for CIFAR-10 when $\epsilon=2/255$. As detailed in section 4.1, also CROWN-IBP carries out convex combinations within the loss, but does so between CROWN-IBP and IBP bounds, and gradually transitions to pure IBP. We refer to section 4.1 for a detailed analysis of the differences between CROWN-IBP and CC-IBP.
>
> > The insight and intuition of the relationship between CC-IBP and MTL-IBP are not clear. For example, does any case exist where one can be degraded to the other?
>
> As shown in proposition 4.2 and described in the relative section, the CC-IBP loss is a lower bound to the MTL-IBP loss. The two losses trivially coincide for $\alpha=0$ and $\alpha=1$, as they both satisfy definition 3.1. The insight we want to convey by presenting two different losses (and also taking into account the SABR results, see Figure 3 and Table 6), is that *expressivity, rather than the specific form of the loss function, leads to state-of-the-art performance*.
>
> > For Table 1, the proposed method is with BaB as a complete method, I wonder if it is fair to compare with some incomplete baselines.
>
> Note that (S)TAPS, SABR, IBP-R and COLT from Table 1 report results under complete verifiers, and report similar comparisons.
> Generally speaking, methods trained exclusively using incomplete verifiers (for instance, IBP, CROWN-IBP) display only very minimal improvements in verified accuracy when moving from incomplete verifiers to complete verifiers at evaluation time.
> Evidence for this can be found in Figure 2, which shows results for IBP training when $\alpha=1$. Importantly, they also produce much lower standard accuracies as shown in Table 1.
> In addition, Table 4 in appendix F.1 shows the verified accuracies of our models under incomplete verifiers. On larger datasets (TinyImagenet, Imagenet64), both MTL-IBP and CC-IBP already attain significant verified accuracies under CROWN-IBP bounds (hence without resorting to complete verifiers).

---

### Author Response · Authors · 2023-11-20
**General Response**

We thank all the reviewers for their time, questions and comments.

We are happy that three out of four reviewers (tVkU, oQBq, eo5p) found our paper to be mostly well-written ("well-organized and easy to follow" - tVkU, "The presentation is mostly good" - oQBq, "generally well written and presented" - eo5p). We are confident that the provided comments will improve the quality of the writing. We are also glad that three reviewers (tVkU, 2HZC, oQBq) positively evaluated our experiments ("the experiments are comprehensive and different datasets and attack radii are used" - tVkU, "The empirical results seem strong" - 2HZC, "Experimental evaluation of CC-IBP and MTL-IBP. Both attain sota" - oQBq).

The main concern with our submission, raised by reviewers tVkU and eo5p, pertains to the *novelty* of the work, evaluated in terms of similarity to previous work (reviewer tVkU points out that SABR satisfies our definition of expressivity, as we explain, and the use of different convex combinations in previous work) or to the simplicity of the idea of using convex combinations ("convexly combining loss functions is a rather obvious idea [...] which has been around for ages" - eo5p). We welcome the opportunity to answer these points.

We are well aware (as discussed in section 4) that convex combinations have been used before in certified training, albeit with different purposes and forms with respect to our submission (cf. section 4). However, rather than a weakness, we believe this to be a strength of our work.
**We show that extremely simple and common techniques result in state-of-the-art performance when they are recast, for the first time, under the lens of our novel notion of expressivity**.
This points to the centrality of definition 3.1 to state-of-the-art certified training, which we believe to be the main contribution of our work, together with the supporting experimental evidence (including the link between performance and approximations of the branch and bound loss).
Indeed, reviewers 2HZC and oQBq contrasted the simplicity of MTL-IBP and CC-IBP with their empirical performance: "The empirical results seem strong, especially considering the fact that the proposed methods are simple interpolations." (2HZC), "The ideas are conceptually simple yet seem to be effective." (oQBq). On TinyImageNet and ImageNet64, MTL-IBP and CC-IBP outperform results from the literature by a sizeable margin in spite of their simplicity.
Our contribution is also well-represented by reviewer oQBq, who wrote that our work "unifies and generalizes successfull approaches".
To paraphrase the title of the SABR paper ("small boxes are all you need"), we aim to show that **expressive losses are all you need**.

We hope that the reviewers can reevaluate our contributions in light of the above, and would be eager to engage in further productive and interesting discussions.

---

### Author Response · Authors · 2023-11-22
**New expressive loss presented**

We would like to thank again all the reviewers for their precious feedback, time, and thoughtful comments, which will significantly strengthen our work.

As a result of the productive and constructive exchange with reviewer eo5p, we experimented with a new expressive loss function, which we provisionally call Exp-IBP, that takes the following form: $$L_{exp} = L_{adv} ^ {1 - \alpha}  L_{ver} ^ {\alpha}.$$
Our experiments on CIFAR $8/255$ (the only benchmark we considered for this, as agreed with reviewer eo5p owing to lack of time before the end of the discussion phase) show that Exp-IBP performs on par with CC-IBP and MTL-IBP (and, transitively, at least as well as the recent state-of-the-art approaches SABR (ICLR23) and STAPS (NeurIPS23)).

| Method    | Source | Std. Acc. | Ver. Rob. Acc. |
| -------- | ------- | -------- | ------- |
| SABR  | Mueller et al. (2023)  |  52.38 | 35.13 |
| STAPS  | Mao et al. (2023a)  |  52.82 | 34.65 |
| CC-IBP  | Table 1 (this work)  |  53.71 | 35.27 |
| MTL-IBP  | Table 1 (this work)  |  53.35 | 35.44 |
| **Exp-IBP**  | new experiment (this work)  |  **56.34** | **34.55** |

We believe that these results strengthen the core message of the paper: our novel definition of expressivity is a key concept in verified training. In particular, it is linked with state-of-the-art performance and to recent advances in the literature, which it generalizes.
Amongst the supporting evidence, we presented two new and extremely simple algorithms that yield state-of-the-art results in a variety of benchmarks (CC-IBP and MTL-IBP), and a third simple expressive loss (Exp-IBP) that matches their performance on the only benchmark we had the time to consider in the last part of the discussion phase. We believe these to be worthy contributions to the community on their own, even besides the notion of expressivity.

We thank the reviewers again for the constructive feedback, and sincerely hope that they can take the above discussion and results into account in their evaluation.

---

### Meta-Review · Area_Chair_bFTL · 2023-12-15

**Metareview:**

This paper introduces a novel approach that combines verified and adversarial loss functions in a convex manner to train models with verifiable robustness. The authors emphasize the importance of the expressivity of the loss function in achieving state-of-the-art performance. The authors effectively addressed the concerns raised by the reviewers during the rebuttal phase.

**Justification For Why Not Higher Score:**

I reached this decision by evaluating the contributions and novelty of the work, taking into consideration both the reviews and the responses from the authors.

**Justification For Why Not Lower Score:**

I reached this decision by evaluating the contributions and novelty of the work, taking into consideration both the reviews and the responses from the authors.

---

### Decision · Program_Chairs · 2024-01-16

Accept (poster)